



# Aerosol pH and chemical regimes of sulfate formation in aerosol water during winter haze in the North China Plain

Wei Tao[1,2], Hang Su[1*], Guangjie Zheng[2], Jiandong Wang[2], Chao Wei[1], Lixia Liu[2], Nan Ma[3], Meng Li[1], Qiang Zhang[4], Ulrich Pöschl[1], Yafang Cheng[2,1*]

[1]Multiphase Chemistry Department, Max Planck Institute for Chemistry, Mainz 55128, Germany
[2]Minerva Research Group, Max Planck Institute for Chemistry, Mainz 55128, Germany
[3]Institute for Environmental and Climate Research, Jinan University, Guangzhou 511443, China
[4]Department of Earth System Science, Tsinghua University, Beijing 100084, China

*Correspondence to*: H. Su (h.su@mpic.de) or Y. Cheng (yafang.cheng@mpic.de)

## Abstract

Understanding the mechanism of haze formation is crucial for the development of deliberate pollution control strategies.

Multiphase chemical reactions in aerosol water have been suggested as an important source of particulate sulfate during severe haze (Cheng et al., 2016;Wang et al., 2016). While the key role of aerosol water has been commonly accepted, the relative importance of different oxidation pathways in the aqueous phase is still under debate, mainly due to questions about aerosol pH. To investigate the spatio-temporal variability of aerosol pH and sulfate formation during winter in the North China Plain (NCP), we have developed a new aerosol water chemistry module (AWAC) for the WRF-Chem model

(Weather Research and Forecasting model coupled with Chemistry). Using the WRF-Chem-AWAC model, we performed a comprehensive survey of the atmospheric conditions characteristic for wintertime in the NCP, focusing on January 2013. We find that aerosol pH exhibited a strong vertical gradient and distinct diurnal cycle, which was closely associated with the spatio-temporal variation in the relative abundance of acidic and alkaline fine particle components and their gaseous counterparts. Over Beijing, the average aerosol pH at the surface layer was ~5.4 and remained nearly constant around ~5

up to ~2 km above ground level; further aloft, the acidity rapidly increased to pH ~0 at ~3 km. The pattern of aerosol acidity increase with altitude persisted over the NCP, while the specific levels and gradients of pH varied between different regions. In the region north of ~41°N, the mean pH values at surface level were typically >6 and the main pathway of sulfate formation in aerosol water was S(IV) oxidation by ozone. South of ~41°N, the mean pH values at surface level were typically in the range of 4.4 to 5.7, and different chemical regimes and reaction pathways of sulfate formation prevailed in

four different regions, depending on reactant concentrations and atmospheric conditions. The $NO_2$ reaction pathway



prevailed in the megacity region of Beijing and the large area of Hebei Province to the south and west of Beijing, as well as part of Shandong Province. The transition metal ion (TMI) pathway dominated in the inland region to the west and the coastal regions to the east of Beijing, and the $H_2O_2$ pathway dominated in the region extending further south (Shandong and Henan Provinces). In all of these regions, the $O_3$ and TMI pathways in aerosol water as well as the gas-particle

5    partitioning of $H_2SO_4$ vapor became more important with increasing altitude. Although pH is sensitive to the abundance of $NH_3$ and crustal particles, we show that the rapid production of sulfate in the NCP can be maintained over a wide range of aerosol acidity (e.g., pH = 4.2-5.7) with transitions from TMI pathway dominated to $NO_2/O_3$ pathway dominated regimes.

**Keywords:** Winter haze, aerosol pH, multiphase chemistry, sulfate formation regime



## 1. Introduction

Persistent haze shrouding Beijing and its surrounding areas in North China Plain during cold winter is one of the most urgent and challenging environmental problems in China (Sun et al., 2014;Zheng et al., 2015b;Cheng et al., 2016). The winter haze often has the following characteristic features, including stagnant meteorological conditions, high relative

humidity and high concentrations of $PM_{2.5}$ as well as elevated contributions of secondary inorganics in $PM_{2.5}$ (Brimblecombe, 2012;Sun et al., 2014;Zhang et al., 2014;Zheng et al., 2015b). Though extremely sharp increases in $PM_{2.5}$ concentration in Beijing (e.g., several hundred $\mu g\ m^{-3}\ h^{-1}$) have been attributed mainly to the transport processes rather than local chemical production, the large gap between modeled and observed $PM_{2.5}$ reveals that there are still missing chemical formation pathways in the state-of-the-art atmospheric chemical transport models (Zheng et al., 2015b;Cheng et al., 2016).

Cheng et al. (2016) suggested and quantified that during severe haze multiphase reactions in aerosol water can produce remarkable amount of sulfate over a wide range of aerosol pH, which complements or even exceeds the contribution from gas phase and cloud chemistry during the haze events. Laboratory studies of Wang et al. (2016) provide an experimental proof for importance of $NO_2$ oxidation pathway in sulfate formation in aerosol water. However, depending on the aerosol pH and pollutant compositions, the major multiphase oxidation pathways may change from reactions of $NO_2$ and $O_3$ at pH >

4.5 to $O_2$ (catalyzed by Transition Metal Ion, TMI) and $H_2O_2$ at pH<4.5 (Cheng et al., 2016). Unlike the negative feedback between aerosol loadings and their photochemical production (Kong et al., 2015;Makar et al., 2015), the multiphase reactions induce a positive feedback mechanism, i.e., higher particle matter levels lead to more aerosol water, which accelerates sulfate production and further increases the aerosol concentration (Cheng et al., 2016).

Though the importance of reactions in aerosol water during severe haze has been widely accepted (Zhang et al., 2015;Zheng

et al., 2015b;Chen et al., 2016;Cheng et al., 2016;Wang et al., 2016;Li et al., 2017a;Chen et al., 2019;Gen et al., 2019;Shao et al., 2019;Wu et al., 2019;Xue et al., 2019), the exact formation pathway is still under debate. Except for the aforementioned reactions (i.e., reactions of $NO_2$, $O_3$, TMI and $H_2O_2$), heterogeneous production of hydroxymethanesulfonate (HMS) by $SO_2$ and HCHO has also been proposed to contribute to the unexplained sulfate by models (Moch et al., 2018;Song et al., 2019). To a certain extent, this is not a surprise considering the strong dependence

of multiphase reaction rate on aerosol pH and pollutant compositions (including the most important oxidants for sulfate formation). Hourly-predicted pH based on observations and thermodynamic model calculations showed a large variability from 2 to 8 in northern China  (Shi et al., 2017;Ding et al., 2019). Previous observational and modeling studies also indicated that the temporal and spatial distribution of oxidants/catalysts, e.g., $O_3$ (Xu et al., 2008;Dufour et al., 2010), $NO_2$ (Zhang et al., 2007;Zhang et al., 2012) and TMI (Dong et al., 2016), were highly variable.



Thus, we hypothesize that multiple oxidation regimes for sulfate formation may indeed co-exist in North China Plain. The apparent contrasting results could be a consequence of regime transitions between different locations and periods. To test our hypothesis, we have developed a new aerosol water chemistry module (AWAC) and implemented an improved version of ISORROPIA II into the WRF-Chem model (Weather Research and Forecasting model coupled with Chemistry) to better

account for the different sulfate formation pathways. With a comprehensive model survey, we focus on the variabilities of aerosol pH and regimes of sulfate formation in aerosol water, aiming at reconciling the continuing debates on the dominate sulfate formation pathways in North China Plain. Detailed method description and model evaluation are provided in Section 2, followed by the results and discussion in Section 3. The last section summarizes the main conclusions and implications of this study.

## 2. Method

### 2.1 WRF-Chem-AWAC: new aerosol water chemistry module for WRF-Chem

To account for the formation of sulfate and nitrate in the liquid water of fine particles (including Aitken and accumulation modes, and denoted as $PM_{2.5}$), we have developed a new aerosol water chemistry module (AWAC) into WRF-Chem (version 3.8) (Grell et al., 2005). The coarse mode aerosols have been greatly simplified in MADE/SORGAM framework

and the AWAC module is therefore not implemented for coarse mode aerosols. For sulfate (denoted as PM25_SO4) formation, we use explicit mechanisms involving 11 irreversible reactions for the oxidation of S(IV) by dissolved $O_3$, $H_2O_2$, TMI (only $Fe^{3+}$ and $Mn^{2+}$ are considered), $NO_2$ and $CH_3OOH$, respectively (Table 1). Mass transfer of $SO_2$ and gaseous oxidants between gas and liquid phase, as well as dissociation equilibrium of sulphurous acid are treated as irreversible reactions, and solved simultaneously with the redox reactions using KPP software (Damian et al., 2002;Sandu and Sander,

2006) and Rosenbrock solver (Shampine, 1982;Sandu et al., 1997). The formula for mass transfer rate coefficient, $k_T$ (in unit of $cm\cdot s^{-1}$) is adopted from Jacob and Brasseur (2017):

$$k_T = (\frac{R_d}{D_g} + \frac{0.04}{\alpha\sqrt{8RT/\pi M_w}})^{-1}$$ (1)

where $R_d$ is the mean radius for fine particles (including aerosol water, with a unit of cm), $D_g$ is the molecular diffusion coefficient ($cm^2\cdot s^{-1}$), $R$ is the gas constant ($8.314$ $J\cdot mol^{-1}\cdot K^{-1}$), $T$ is the air temperature (K), $M_w$ is the molecular weight

($kg\cdot mol^{-1}$), and $\alpha$ is the mass accommodation coefficient (unitless). Detailed descriptions are provided in the Supplement (Fig. S1-S5 and Tables S1-S5). Though using different solvers, the box model version of the aerosol water chemistry module could well reproduce the results in Cheng et al. (2016) (shown in Fig. S5 of Supplement).





Following Zheng et al. (2015a) and Chen et al. (2016), a parameterization scheme is adopted to simulate the aqueous phase production of nitrate (denoted as PM25_NO3):

$$\frac{dC_{\text{PM25\_NO3}}}{dt} = \left( \frac{R_d}{D_{g,\text{NO}_2}} + \frac{0.04}{\gamma_{\text{NO}_2}\sqrt{8RT / \pi M_{w,\text{NO}_2}}} \right)^{-1} \cdot Aa \cdot C_{\text{NO}_2(g)} \tag{2}$$

where $Aa$ is the surface area concentration for fine particles ($cm^2 \cdot cm^{-3}$), and $\gamma$ is the uptake coefficient (unitless) for $NO_2$.

$\gamma$ has a lower limit ($3.0 \times 10^{-6}$) and a higher limit ($13.0 \times 10^{-6}$) when the RH is lower than 50 % and higher than 90 %, respectively. Note that the $\gamma$ has been scaled 15 times lower than that used in Zheng et al. (2015a) to better match the observed PM25_NO3 with R = 0.74 and NMB = 3% (see more details in Sect. 3.1). When the RH ranges between 50 % and 90 %, the value for $\gamma$ is then linearly interpolated between the two limits.

To simulate aerosol water content (AWC) and pH, as input parameters of the AWAC module, we have replaced the original

ISORROPIA model in WRF-Chem with an improved version of ISORROPIA II (Fountoukis and Nenes, 2007;Song et al., 2018). The source code of the improved ISORROPIA II is available at http://wiki.seas.harvard.edu/geos-chem/index.php/ISORROPIA_II. ISORROPIA predicts the thermodynamic equilibrium (including pH) for an internal-mixed system of multiple inorganic components at a specific temperature and humidity. Following Ding et al. (2019), we assume that the phase state for aerosol is metastable at RH > 30%, otherwise the phase state is stable. If the predicted AWC

is less than an infinitesimal value (the threshold value of $10^{-8}$ μg/m$^3$ is used, see Fig. S6 and S7 of Supplement for relevant uncertainties), pH is set to a missing value, and heterogeneous reactions in aerosol water are not calculated. Otherwise the heterogeneous oxidation module is called, assuming a fixed number/size distribution and thermodynamic status (including AWC and pH) within one time step of master chemistry module in WRF-Chem program.

RACM mechanism (Stockwell et al., 1990;Stockwell et al., 1997) is used to calculate the gas phase photochemistry, and

provides the concentrations of gaseous precursors (e.g., $SO_2$, $NO_2$ etc.) and oxidants (e.g., $O_3$, $H_2O_2$ etc.). MADE/SORGAM scheme (Ackermann et al., 1998;Schell et al., 2001) with the improved ISORROPIA II model is used to simulate the aerosol dynamics (including nucleation, coagulation and condensation) and thermodynamics, and provide the aerosol size distribution, number concentration, as well as AWC and pH values. The Integrated process rate (IPR) technique (Tao et al., 2015;Tao et al., 2017) is used to record the formation rates of sulfuric and nitric acid vapor through photochemical oxidation

in the gas phase.

For the TMI oxidation pathway, we assume that $Fe^{3+}$ and $Mn^{2+}$ will not be consumed in the TMI pathway due to the catalytic nature of $Fe^{3+}/Mn^{2+}$. The concentrations of $Fe^{3+}$ and $Mn^{2+}$ (in unit of mol/L) in aerosol water can be calculated by Eq. (3),





$$\begin{cases} [\text{Fe}^{3+}] = Min(C_{\text{PM25\_FE}} \cdot FS_{\text{FE3+}} / AWC, 2.6 \times 10^{-38} / [\text{OH}^-]^3) \\ [\text{Mn}^{2+}] = Min(C_{\text{PM25\_MN}} \cdot FS_{\text{MN2+}} / AWC, 1.6 \times 10^{-13} / [\text{OH}^-]^2) \end{cases} \qquad (3)$$

where PM25_FE and PM25_MN are the fine particle components of Fe and Mn minerals, respectively, $C$ denotes the concentrations in unit of mol per liter of air, $FS_{\text{FE3+}}$ and $FS_{\text{MN2+}}$ represent the maximum fractional solubility of $Fe^{3+}$ and $Mn^{2+}$, respectively (regardless of the acidity of aerosol water), and $AWC$ is the aerosol liquid water content in unit of liter
per liter of air.

### 2.2 WRF-Chem model configuration and scenarios

The modeling framework is constructed on a single domain of 100 (west-east) × 70 (south-north) × 30 (vertical layers) grid cells with a horizontal resolution of 20 km (including the Gobi deserts, see Fig. S8 of Supplement). The overview of the chemical and physical options used in this study is summarized in Table S6 of Supplement. Madronich F-TUV photolysis
scheme (Madronich, 1987) is used to calculate the photolysis rates. The initial and boundary conditions for meteorology and chemistry are derived from 1.0°×1.0° NCEP FNL data and global-scale MOZART outputs, respectively. Observation nudging (Liu et al., 2005) is used to nudge the modeled temperature, wind fields and humidity towards the observations (including surface and upper layers). Multi-resolution Emission Inventory for China (MEIC) of the year 2013 (MEIC, Zhang et al., 2009;Lei et al., 2011;Li et al., 2014;Li et al., 2017b) is used for anthropogenic emissions. The Megan scheme (Guenther
et al., 2006) is used for biogenic VOCs emissions. The hourly biomass burning emissions data are provided by the Fire Inventory from NCAR (FINN, Wiedinmyer et al., 2010). We use the GOCART dust scheme (Ginoux et al., 2001;Zhao et al., 2010;Zhao et al., 2013), which is coupled with the MADE/SORGAM aerosol scheme.

In this study, we have simulated 13 scenarios as detailed in Table 2, including an ORIG scenario with WRF-Chem (original chemistry) and default emissions as mentioned above, a CTRL scenario with WRF-Chem-AWAC (with implementation of
aerosol water aqueous phase chemistry) and optimized ammonia emission, as well as additional emissions for chloride and crustal fine particles, and 11 sensitivity scenarios with respect to CTRL scenario. Since anthropogenic source chlorine is not included in MEIC, we adopt the chlorine inventory in Liu et al. (2018), which provides the emissions of HCl from coal consumption. We also adopt the dust emission speciation profiles from Dong et al. (2016), and the mass fraction for fine particle components of K, Na, Ca, Mg, Fe and Mn minerals (denoted as PM25_K, PM25_NA, PM25_CA, PM25_MG,
PM25_FE and PM25_MN, respectively) from dust source are set as 3.77%, 3.94%, 7.94%, 0.80%, 2.43%, and 0.063%, respectively. Dry and wet depositions are also considered for these newly-added crustal fine particle components as detailed in Sect. 1.3 of Supplement.





The CTRL scenario is expected to reproduce the observed fine particle components and gas phase pollutants, and thus more reliably predict the spatio-temporal distribution of pH and AWC. For this purpose, five parameters in the CTRL scenario have been adjusted to better match the observations (with the criteria that the relative error of the monthly mean concentrations for different fine particle components is less than 5%), namely the factor multiplied to the anthropogenic ammonia emissions

(denoted as $emissf_{NH3}$), the factor multiplied to dust speciation fractions for PM25_K, PM25_NA, PM_CA and PM_MG (denoted as $emissf_{OCAT}$), the factor multiplied to the anthropogenic chloride emissions (denoted as $emissf_{CL}$), as well as $FS_{FE3+}$ and $FS_{MN2+}$ (the maximum fractional solubility of $Fe^{3+}$ and $Mn^{2+}$ in Eq. 3, respectively). In the CTRL scenario, $emissf_{NH3}$ has been set to 2, as recent studies using top-down inverse modeling (Van Damme et al., 2018;Wang et al., 2018;Zhang et al., 2018a;Kong et al., 2019) and direct measurement (Wang et al., 2018) found that previous bottom-up inventories might

underestimate the $NH_3$ emissions, and MEIC inventory was estimated to under-predict $NH_3$ emissions by about 40% over the North China Plain (Kong et al., 2019). As shown in Table S7 of Supplement, doubling the $NH_3$ emissions better match the observed ammonia and ammonium at urban Beijing sites during wintertime (Meng et al., 2011;Liu et al., 2017;Song et al., 2018). To match the observations of PM25_OCAT and PM25_CL, the $emissf_{OCAT}$ and $emissf_{CL}$ are set to 4.5 and 6, respectively. To have a better agreement with the sulfate observation, $FS_{FE3+}$ and $FS_{MN2+}$ are set to 7% and 40%, respectively.

Note that the assumption behind tuning only $FS_{FE3+}$ and $FS_{MN2+}$ to better agree with observed sulfates, is that the model could reasonably simulate the concentrations for other oxidants (e.g., OH, $H_2O_2$, $O_3$ and $NO_2$), thus the deviation from observation can be attributed to the uncertainties in representation of TMI pathway. Journet et al. (2008) found that the dust mineralogy was a critical factor for iron solubility, and fractional Fe solubility was observed to be ~4% and less than 1% for samples of clay and iron (hydr-)oxides, respectively. A low fractional Fe solubility approximately or below 0.05% was reported for the

non-atmospherically-processed arid soil samples (Schroth et al., 2009;Johnson et al., 2010;Shi et al., 2012). Most of Fe minerals exist in the form of hematite (α-$Fe_2O_3$) over the Gobi deserts (Claquin et al., 1999), and fractional Fe solubility was increased to 1-2% after 3–5 days transport from the Gobi deserts (Meskhidze et al., 2003). A higher fractional Fe solubility up to ~10% was reported after a longer time of atmospheric aging for dust particles (Takahashi et al., 2011;Shi et al., 2012). Fractional Mn solubility for dust particles was observed to range between 20% and 60% (Baker et al., 2006;Duvall et al.,

2008;Hsu et al., 2010). The rest 11 scenarios are used for sensitivity analysis on the uncertainties in $emissf_{NH3}$, $emissf_{OCAT}$, $emissf_{CL}$, $FS_{FE3+}$ and $FS_{MN2+}$.



## 3. Results and discussion

### 3.1 Comparison of CTRL and ORIG scenarios

Figure 1 shows the comparison of ORIG and CTRL scenarios against observations obtained at the site located on the campus of Tsinghua University in Beijing (denoted as Beijing site, 40°00'17"N, 116°19'34"E, ~10m height) during January 2013. The hourly data include online observations of concentrations of $SO_2$, PM25_SO4, $NO_x$ (=NO+NO$_2$), PM25_NO3, $O_3$ and $PM_{2.5}$, as well as relative humidity (RH) that is closely related to the AWC. Details about the measurements have been described in Zheng et al. (2015b) and Cheng et al. (2016). As shown in Fig. 1, simulations from both scenarios were capable of reproducing the concentration levels of the sum of $SO_2$ and PM25_SO4, the sum of $NO_x$ and PM25_NO3, $O_3$ and $PM_{2.5}$ with the normalized mean bias (NMB) around ±30% and the correlation coefficients (R) ranging between 0.5 and 0.7. The simulated RH well matched the observations with a NMB of ~12% and a R of ~0.9. Simulations of PM25_SO4, PM25_NO3 have been significantly improved in CTRL scenario.

Offline observations of PM25_K, PM25_NA, PM25_CA, PM25_MG, PM25_CL, PM25_SO4, PM25_NO3 and PM25_NH4 are also available in a daily basis (Zheng et al., 2015b;Cheng et al., 2016). As shown in Fig. 2, simulations of sulfate, nitrate and ammonium in $PM_{2.5}$ were greatly improved in the CTRL scenario, with a NMB within ±5%, while the NMB of the ORIG scenario was -90% for sulfate, -35% for nitrate and -65% for ammonium, respectively. The only source of PM25_CL in ORIG scenario was sea salt emissions, and it was negligible. The improved model WRF-Chem-AWAC with additional chloride emissions from coal combustion could capture the observed PM25_CL with a NMB of only ~10%. Other inorganic cations in $PM_{2.5}$ (denoted as PM25_OCAT, and only involves PM25_K, PM25_NA, PM25_CA and PM25_MG) were not included in original MADE/SORGAM scheme, and thus their concentrations were zero in ORIG scenario. The CTRL scenario could moderately reproduce the concentrations of PM25_OCAT with a NMB of ~60%. Note that, as described in Sect. 2.2, the emission of PM25_OCAT has been adjusted based on the criterial of reducing the mismatch of simulation and observation within 5% on a monthly mean basis, and here the NMB here was calculated based on daily values.

As shown in the lower panel of Fig. 2, among the fine particle components mentioned above, observed PM25_NH4 and PM25_OCAT accounted for about 75% and 25% of the monthly mean ionic charge budget for cations, respectively. Observed PM25_CL, PM25_NO3 and PM25_SO4 accounted for about 20%, 32% and 48% of the monthly mean ionic charge budget for anions, respectively. The CTRL scenario could well capture the observed ionic charge budget, except that the mole fraction of PM25_CA in PM25_OCAT was slightly overestimated (PM25_CA had a higher mass fraction for dust emission speciation), but still much better than in the ORIG scenario. Thus, in the following sections, the CTRL





configuration is used to study the temporal and spatial distribution of aerosol pH and the regime transition of sulfate formation in aerosol water during winter haze events in North China Plain.

### 3.2 Vertical profile of pH over Beijing

Figure 3a shows the vertical profile of the simulated aerosol pH over Beijing site. The mean pH at surface layer was ~5.4
(daytime mean pH ~5.2 and nighttime mean pH ~5.6) and remained nearly constant around ~5 up to ~2 km above ground level (AGL). While above ~2 km AGL, the mean pH exhibited a more rapid decrease and became highly acidic (mean pH ~0) at ~3 km AGL. Based on observation of aerosol compositions, Guo et al. (2016) also reported that pH at middle troposphere of 5 km AGL was highly acidic (mean pH ~-0.7) and was about 1.7 unit lower than at surface layer over the northeastern US. These spatial features for pH were closely associated with the spatio-temporal variation in the relative
abundance of acidic and alkaline fine particle components and their gaseous counterparts. Here, the acidic fine particle components include total sulfate ($SO4^T = H_2SO_4(g) + PM25\_SO4$), total nitrate ($NO3^T = HNO_3(g) + PM25\_NO3$) and total chloride ($Cl^T = HCl(g) + PM25\_CL$), and the alkaline components include total ammonia ($NH_x=NH_3(g)+PM25\_NH4$) and PM25_OCAT.

Thus, we define the concentrations for total potential anion (anion$^T$) and total potential cation (cation$^T$) as:

$$C_{anion^T} = 2 \cdot C_{SO4^T} + C_{NO3^T} + C_{Cl^T} \tag{4}$$

$$C_{cation^T} = C_{NH_x} + C_{PM25\_NA} + C_{PM25\_K} + 2 \cdot C_{PM25\_MG} + 2 \cdot C_{PM25\_CA} \tag{5}$$

As shown in Fig. 3b-3c, the concentration ratio of cation$^T$ to anion$^T$ slightly changed (consistently decreased with the increasing height in night-time, but firstly decreased and then increased with the increasing height in daytime) below ~2 km AGL, and rapidly decreased above ~ 2 km AGL. NH$_x$ had the predominant mole fraction in the sum of anion$^T$ and
cation$^T$ below the lower free troposphere (below ~1 km AGL), but its concentrations decreased sharply with the increasing height (Fig. 3e) as ammonia only had surface emissions. Although also dominated by the surface emission source of SO$_2$ and NO$_x$, both $SO4^T (= H_2SO_4(g) + PM25\_SO4)$ and $NO3^T (= HNO_3(g) + PM25\_NO3)$ could be produced through different gas-phase and aqueous-phase oxidation pathways at different altitudes. Thus, although their concentrations also decreased with the increasing altitude (Fig. 3d), the decreasing rate was slower than that of NH$_x$ (Figs. 3d-3e), which led to an increase
in mole fraction of anion$^T$ above ~ 2 km AGL (Figs. 3b-3c). The vertical profile of PM25_OCAT was distinct from that of NH$_x$ (Fig. 3e), with its concentrations remained almost constant until ~1.5 km AGL and relatively slowly decreased above it, suggesting a different source possibly from high-altitude transport. Based on satellite, lidar and surface measurement


data, Huang et al. (2008) studied the vertical structure of Asian dust originated from Taklamakan and Gobi deserts, and found that dust particles could be uplifted to an altitude of ~ 9 km around the source region, followed by the efficient eastward transport mainly via westerly jets. Zhang et al. (2018b) also investigated the dust layering structure over some cities in East Asia (including Beijing, Seoul and Tokyo), and indicated that the dust particles were well mixed in the

boundary layer before the intensive intermingling of subsiding layers from above (the passage of dust storm). Our simulation also shows that PM25_OCAT over the Beijing site was evenly mixed after a long-range transport from its source region (mainly in Gobi deserts, shown in Fig. S9 of Supplement) below the lower free troposphere. Under different environmental conditions, the changes in concentrations and mole fractions of acidic and alkaline components lead to competition and different dominator factors in the changes of pH vertically. Above ~2km, the influences of acidic

components (mainly $SO4^T$ and $NO3^T$) on pH eventually prevailed over those from alkaline components. Interestingly, PM25_OCAT could play a key role in maintaining a less acidic pH at ~1.5-2.0 km AGL. Sensitivity test shows that if the PM25_OCAT is completely removed (in OCAT0 scenario), pH at 2.0 km AGL would be ~ 1 unit lower (more acidic) than that at 1.5 km AGL.

### 3.3 Diurnal cycle of pH over Beijing

Figure 3a shows that at the Beijing site the night-time pH was slightly higher than the daytime pH below ~1 km AGL, but became lower above ~ 1 km. As shown in Fig. 4, such opposite pattern was largely driven by a different diurnal cycle of aerosol pH at different altitudes. Aerosol pH at surface layer showed a minimum in the early afternoon while at 2 km it showed a maximum around this time (Fig. 4a-4b). The diurnal variation of pH also showed a highly similar pattern as the mole ratio of cation$^T$ to anion$^T$ (Fig. 4a-4b). At surface layer, concentrations of cation$^T$ were considerably higher than

concentrations of anion$^T$ (Fig. 4d), furthermore the diurnal variation of $NH_x$ to a large extent explained the diurnal cycle pattern of cation$^T$. Daytime $NH_x$ concentrations were significantly lower than at the night-time, due to a stronger boundary layer mixing. Following Song et al. (2018), we define the concentrations of required $NH_x$ ($NH_x^{Req}$) as:

$$C_{NH_x^{Req}} = 2 \cdot C_{SO_4^T} + C_{NO_3^T} + C_{Cl^T} - C_{PM25\_NA} - C_{PM25\_K} - 2 \cdot C_{PM25\_MG} - 2 \cdot C_{PM25\_CA} \qquad (6)$$

The physical meaning for $NH_x^{Req}$ is the minimum $NH_x$ ideally immobilizes all the gas phase $H_2SO_4$, $HNO_3$ and $HCl$. Thus,

the concentration ratio of $NH_x^{Req}$ to $NH_x$ describes the levels of ammonia excess (Liu et al., 2017;Song et al., 2018). Diurnal cycle of pH and ammonia excess matched quite well with each other (Fig. 4b), and enhanced ammonia excess corresponded with a higher pH during night-time. PM25_OCAT (mainly originated from long-range transport) concentrations were higher in daytime. $SO4^T$ rapidly accumulated during the night (when the aqueous phase production was active), while $NO3^T$ concentrations reached the peak in the late afternoon (gas phase oxidation played a more important role). As shown



in Fig. 4c, a higher aerosol water content during night-time (due to higher RH and PM$_{2.5}$ loading) would also contribute to the less acidic condition. At layer of ~2 km AGL, pH diurnal cycle was also similarly associated with the variation in mole ratio of cation$^T$ to anion$^T$ (peaks around the noon), and the diurnal variation of PM25_OCAT might play an important role.

### 3.4 pH variabilities over the North China Plain

Figures 5a-5b show that aerosol pH at surface layer exhibited a large spatial variability in the North China Plain. High pH > 6 was found in areas north of ~41°N, which could be attributed to abundant crustal components originated from dust and low concentrations of other aerosol inorganic compositions due to low emissions of precursors (e.g., SO$_2$, NO$_x$ and NH$_3$). In areas south of ~41°N, mean aerosol pH fell mostly between 4.4 and 5.7 (10% and 90% quantiles, respectively) with less contribution of crustal components. Mean pH over the sea mainly ranged between 4.0 and 4.5, generally lower than over

the most terrestrial areas. These marine and terrestrial areas with a relatively more acidic aerosol phase (pH ~4.0 to 5.0) corresponded well with the spatial distribution of low NH$_x$ zones (Fig. 5d). Besides, a large temporal variability of pH was also found at the surface layer (Fig. 5c). The simulated standard deviation of pH mostly ranged between 0.4 and 2 and was higher over the northern areas with episodic dust events, as well as the southern areas with lower NH$_x$ emissions.

Figure 6a shows the latitude-height cross section of aerosol pH in the North China Plain. The trend that aerosol acidity was

enhanced with the increasing altitudes was consistent for all the latitudes investigated. Nonetheless, the vertical gradients of pH varied among different locations, closely associated with the vertical profile of the relative abundance of cation$^T$ and anion$^T$, as well as air temperature and humidity (Fig. S10 of Supplement). Figure 6b further compares the vertical profiles of aerosol pH over seven different cities in the North China Plain (locations shown in Fig. 5c). The vertical profile pattern of aerosol pH was highly similar among Beijing, Tianjin, Baoding and Shijiazhuang, wherein the pH changed slightly until

above ~ 2 km AGL. However, as shown in Fig. 5c and Fig. S10, pH decreased rapidly from 1 to 2 km AGL over both Zhangjiakou and Taiyuan, maybe due to the lack of alkaline fine particle components (NH$_x$ or PM25_OCAT). Rapid production of sulfate and nitrate at 0.5-1 km AGL over Jinan was observed (Fig. S10), leading to a lower pH there. Furthermore, monthly mean night-time pH was mostly higher than daytime pH in the lowest boundary layer below ~100 m AGL, however no consistent pattern was found for the diurnal cycle pattern of pH in the upper tropospheric layers (Fig.

6c).

### 3.5 Regime transition of sulfate formation

Figure 7a shows the averaged contribution of six sulfate formation pathways, namely aerosol water phase oxidation by dissolved O$_3$, NO$_2$, H$_2$O$_2$, TMI (in the presence of O$_2$) and CH$_3$OOH, as well as gas-particle partitioning of H$_2$SO$_4$ vapor (GPP) at the surface layer. Note that H$_2$SO$_4$ vapor is produced mainly from the oxidation of SO$_2$ by OH radical, and then





partitions almost completely into the aerosol phase. For areas north of 41°N, oxidation by dissolved $O_3$ was the most important pathway, followed by TMI pathway. To the south of 41°N, $NO_2$, TMI and $H_2O_2$ pathways played the dominant role. The $NO_2$ reaction pathway prevailed in the megacity region of Beijing and the large area of Hebei Province to the south and west of Beijing, as well as part of Shandong Province, while the TMI pathway dominated in the inland region to the west and the coastal regions to the east of Beijing, and the $H_2O_2$ pathway dominated in the region further south in Shandong and Henan provinces. The regime transition of sulfate formation pathways highly depended on the spatial distribution of pH and oxidants/catalysts. As shown in Fig. 7b-7e, spatial distribution for $O_3$ exhibited an opposite pattern as $NO_x$, i.e., $O_3$ concentrations in urbanized areas (e.g., Beijing) with high $NO_x$ emissions were considerably lower than less industrialized areas, due to the titration effects of NO.

Figure 8b shows the vertical profile of sulfate production rates averaged over the four dominant sulfate formation regimes at the surface layer (i.e., $O_3$, $NO_2$, TMI and $H_2O_2$, shown in Fig. 8a). Consistent with the surface layer, the vertical regime transition of sulfate formation highly depended on the vertical distribution of pH and oxidants/catalysts (Fig. 8c-8f). Both $O_3$ and TMI pathways played the dominant role at higher altitudes between 0.5-2.0 km AGL (Fig. 8b). Interestingly, the vertical profile pattern for pH potentially enhanced TMI pathway but hindered $O_3$ pathway, while the vertical profile pattern for concentrations of oxidants/catalysts disfavored TMI pathway but favored $O_3$ pathway. As shown in Fig. 8b, the relative contributions of $NO_2$ pathway rapidly decreased with the increasing altitudes, consistent with the decreasing trend in both $NO_2$ concentrations (Fig. 8f) and aerosol pH. $H_2O_2$ pathway was non-negligible only in the lower boundary layer over the $H_2O_2$-regime and $NO_2$-regime. GPP pathway tended to become more important with the increasing altitudes, mainly due to the decreasing trend in aerosol water content. At higher altitudes above ~3 km AGL, aqueous phase oxidation in aerosol water became negligible.

Compared to nitrate, aqueous phase oxidation was more important to sulfate formation in North China Plain. For example, according to our simulation, over the Beijing site, below ~2 km AGL, aqueous phase oxidation in aerosol water accounted for the ~100% and 80-90% of $SO_4^T$ formation during night-time and daytime, respectively. While, its respective contributions to night-time and daytime $NO_3^T$ formation were only 40-70% and 0-10%.

## 3.6 Discussion

The uncertainties of predicted pH and sulfate formation relevant to the adjusted emission parameters (for $NH_3$, crustal minerals, $Fe^{3+}/Mn^{2+}$ ions and chlorides) and assumed phase state are further investigated. We focus on 4 aspects, namely surface layer pH (denoted as $pH_{surf}$) and its day/night difference (denoted as $\Delta pH_{surf,night-day}$), the difference between surface layer pH and pH at ~2 km AGL (denoted as $\Delta pH_{surf-2km}$), and surface layer sulfate production rates (denoted as $P_{S(VI),surf}$)



through different pathways for the domain-wide grid cells south of 41°N. As shown in Fig. 9, different scenarios predict different $pH_{surf}$ and $P_{S(VI),surf}$, nonetheless $\Delta pH_{surf,night-day}$ and $\Delta pH_{surf-2km}$ is approximately 0.2 and 5.0, respectively for almost all the sensitivity tests, indicating that the diurnal cycle pattern at surface layer and the altitudinal decrease in pH shall be robust.

We first examine the sensitivities of our results to $NH_3$ and crustal minerals. When $NH_3$ emissions are completely removed (A0 scenario), a strong acidic aerosol phase is predicted (mean $pH_{surf}$ ~0), meanwhile the $P_{S(VI),surf}$ decreases by ~60% as in CTRL scenario (due to lack of efficient water-absorbing ammoniates). In this case, the high concentrations of $H^+$ make the aerosol pH less sensitive to addition of acidic/basic compounds. If PM25_OCAT is removed (OCAT0 scenario), averaged pH decreases to 4.2, and $P_{S(VI),surf}$ increases by ~250% compared to the CTRL scenario (a lower pH favors sulfate
production through TMI pathway). Doubling both $NH_3$ and PM25_OCAT emissions (A2 and OCAT2 scenarios) leads to a slightly higher $pH_{surf}$ and a similar $P_{S(VI),surf}$, with the sulfate formation dominated by $NO_2$ and $O_3$ pathways. Using the original MEIC inventory for $NH_3$ emissions (MEIC_CTRL scenario) predicts a lower $pH_{surf}$ (4.5±0.7) and a similar sulfate production budget as in the CTRL scenario. Interestingly, rapid production of sulfate could be maintained over a wide pH range (~4.2-5.7) with the varying emissions for $NH_3$ and crustal particles (transition between TMI pathway dominated and
$NO_2/O_3$ pathway dominated).

We have also investigated the effect of emissions of chlorides and $Fe^{3+}/Mn^{2+}$ ions. Removing all the chloride emissions (CL0 scenario) has a negligible effect on both aerosol pH and sulfate production. However, if the chloride emissions are doubled (CL2 scenario), pH slightly decreases to 5.0, and $P_{S(VI),surf}$ increases by 50% (maybe increase in $NH_4Cl$ leading to an enhanced water absorption). When both $FS_{FE3+}$ and $FS_{MN2+}$ equal to zero (TMI0 scenario, and TMI pathway is shut
down), $P_{S(VI),surf}$ decrease almost by half and $pH_{surf}$ (5.5) is slightly higher. When both $FS_{FE3+}$ and $FS_{MN2+}$ are doubled (TMI2 scenario), $P_{S(VI),surf}$ increase by ~300% and $pH_{surf}$ decreases to 4.6. Our results indicate that sulfate production is rather sensitive to the availability of TMI ions. Unfortunately, the concentrations as well as sources for $Fe^{3+}/Mn^{2+}$ ions in aerosol water during haze episodes remain not well constrained and understood. Previous observational studies reported that the concentrations for $Fe^{3+}/Mn^{2+}$ ions in cloud water over China was several μM (Guo et al., 2012;Shen et al., 2012;He
et al., 2018), considerably smaller than predicted in this study (0.2 mM for $Fe^{3+}$ and 8 mM for $Mn^{2+}$ in aerosol water). $Fe^{3+}/Mn^{2+}$ ions also have anthropogenic source, and were estimated to account for 10-30% in Beijing (Shao et al., 2019). Furthermore, the soluble Fe/Mn speciation (including $Fe^{3+}$-$Fe^{2+}$, $Mn^{2+}$-$Mn^{3+}$-$Mn^{4+}$ cycling) depends on dust mineralogy, particle acidity and heterogeneous redox reactions (Schroth et al., 2009;Takahashi et al., 2011), and is very difficult to be explicitly treated. Also the activity coefficients for $Fe^{3+}/Mn^{2+}$ ions under the high ionic strength environment might differ
(Cheng et al., 2016). The treatment of TMI pathway should be further improved in future studies.



Different phase state assumptions predict slightly different $pH_{surf}$, but distinct sulfate production. Compared with CTRL scenario, assuming a fixed metastable phase state (MSTB scenario) predicts a slightly lower $pH_{surf}$ (5.0) and 40% higher $P_{S(VI),surf}$, and the contribution of TMI pathway increases. Assuming a fixed stable phase state predicts a slightly higher pH (5.3) and 25% lower $P_{S(VI),surf}$ (maybe mainly due to the changes in predicted aerosol water content), and the contribution of $NO_2$ pathway increases. Previous box model studies reported a similar finding regarding the minor impacts of phase state assumption on pH (Song et al., 2018).

## 4. Conclusions

The focus of the current study is to investigate the spatio-temporal variabilities of aerosol pH and regime transitions of sulfate formation at a regional scale. For this purpose, an aqueous phase chemistry module (AWAC) for sulfate and nitrate formation in aerosol water has been developed and implemented, using the results (including aerosol water and pH) of revised ISORROPIA II model as input data. With this improved version of Weather Research Forecasting Model with Chemistry (WRF−Chem), we simulated the severe and successive haze pollution spreading over the North China Plain during January of 2013. Control experiments could well reproduce the observed inorganic components of fine particles (including sulfate, nitrate, ammonium, chloride, sodium, and crustal minerals), as well as $SO_2$, $NO_2$, $O_3$ and relative humidity, compared to the observations at the Beijing Tsinghua University site. Impacts of the uncertainties in parameters and assumptions have also been discussed.

The vertical profile and diurnal cycle pattern for aerosol pH were closely associated with the spatio-temporal variation in the relative abundance of acidic and alkaline fine particle components and their gaseous counterparts. The competition between the ammonia, crustal particles and acidic components (such as sulfate and nitrate) could play an important role in determining pH in different vertical layers. The monthly mean pH at surface layer exhibited a large spatial variability over the North China Plain. Mean pH was greater than 6 for the areas with higher latitudes north of 41°N (mainly influenced by the abundant crustal particles) and was mostly within 4.4 and 5.7 (10% and 90% quantiles, respectively) for vast areas south of ~41°N (with $NH_x$ as the driving factor) over the North China Plain. The trend that aerosol acidity was enhanced with the increasing altitudes was consistent for all latitudes (35-43°N) investigated, while the vertical gradients of pH varied between different locations. The diurnal cycle pattern existed only for the domain-wide cells in the lower boundary layer, wherein night-time pH was higher.

In the AWAC module, six sulfate formation pathways in aerosol water are implemented and compared, namely aqueous phase oxidation by dissolved $O_3$, $NO_2$, $H_2O_2$, TMI (in the presence of $O_2$) and $CH_3OOH$, as well as gas-particle partitioning of $H_2SO_4$ vapor (GPP). The relative contributions of different sulfate formation pathways in aerosol water depended on



both pH as well as the concentrations of each oxidant/catalyst. At surface layer, $O_3$, $NO_2$, TMI and $H_2O_2$ pathways were the most important in different locations over the North China Plain, and four regions with three distinct regimes have been found. With the increasing height, $O_3$, TMI and GPP pathways became more important, while contributions from $NO_2$ and $H_2O_2$ pathways decreased rapidly. At higher altitudes above ~3 km above ground level, aqueous phase oxidation in aerosol

water became negligible.

The diurnal cycle pattern at surface layer and altitudinal decrease for pH is consistent for all the sensitivity tests with varying adjusted emission parameters and phase state assumptions except when $NH_3$ emissions are completely removed. pH is sensitive to the alkalization effect of $NH_3$ and crustal particles, furthermore, rapid production of sulfate could be maintained over a wide pH range (e.g., 4.2-5.7) with the varying emissions for $NH_3$ and crustal particles (transition from

TMI pathway dominated to $NO_2$/$O_3$ pathway dominated). The sulfate production is rather sensitive to the concentrations of TMI ions and doubling the TMI ions sources almost triples the sulfate production. Changes in chloride emissions as well as phase state assumptions both have a relatively minor effect on pH, but sulfate formation could be changed as the predicted aerosol water changes. Our studies suggest that sources of crustal particles, $NH_3$ and TMI ions are very important factors for the aqueous phase chemistry during haze episodes and should be better constrained in future studies. Moreover,

the use of a more detailed aqueous phase mechanism involving the TMI ions cycling and radical chain propagation is suggested. Impacts of size distribution, mixing state and organic matters on the aqueous phase chemistry, as well as the contribution from the heterogeneous production of HMS by $SO_2$ and HCHO, should also be addressed in future studies.

### Acknowledgements

This study is supported by Max Planck Society (MPG). Y. Cheng would like to thank the Minerva program of MPG.

**Author contribution**: H.S. and Y.C. designed and led the study. W.T. developed the AWAC module and implemented the revised ISORROPIA II into WRF-Chem. W.T. performed model simulations. W.T., H.S. and Y.C. analyzed data and interpreted the results. G.Z. supported the data analyses. J.W., C.W. and L.L. supported modeling work. M.L. and Q.Z. provided the MEIC for the year of 2013. All coauthors have discussed results and commented on the manuscript. W.T. wrote the manuscript with input from all coauthors.





## Figures

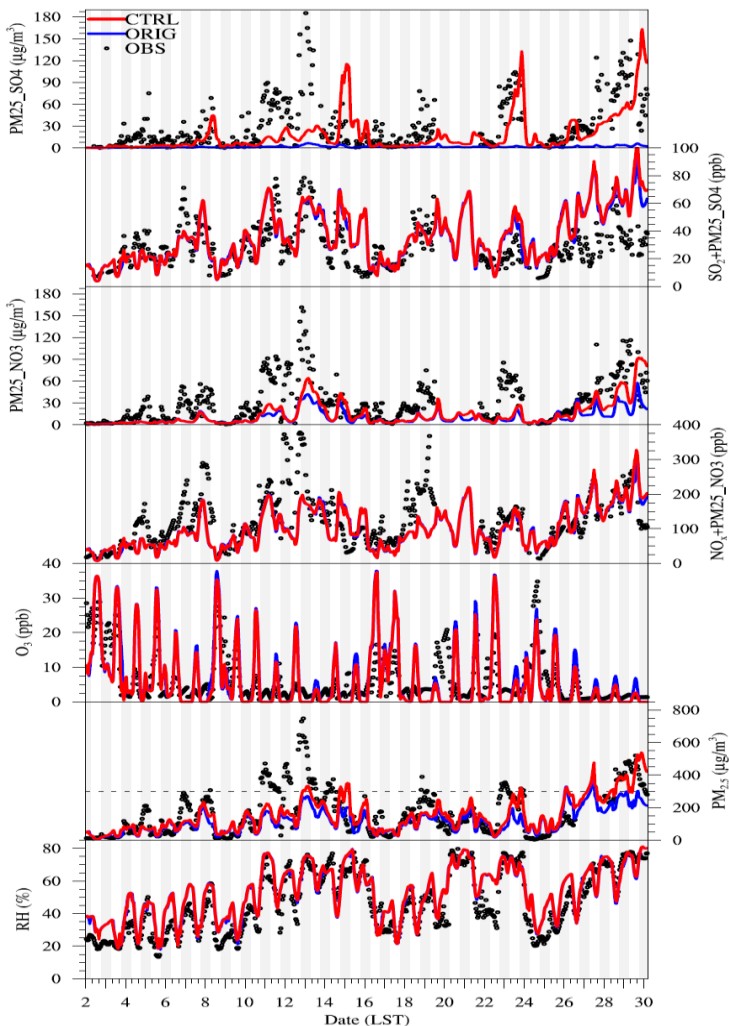

Figure 1. Comparison of hourly model results and observations (OBS) for ORIG and CTRL scenarios at Beijing TSU site during January 2013. The compared parameters include modelled hourly concentrations of fine particulate sulfate (PM25_SO4), the sum of $SO_2$ and PM25_SO4 (PM25_SO4 converted to the equivalent volume mixing ratio of $SO_2$), fine particulate nitrate (PM25_NO3), the sum of $NO_x$ and PM25_NO3 (PM25_ NO3 converted to the equivalent volume mixing ratio of $NO_x$), $O_3$ and $PM_{2.5}$, as well as hourly relative humidity (RH).

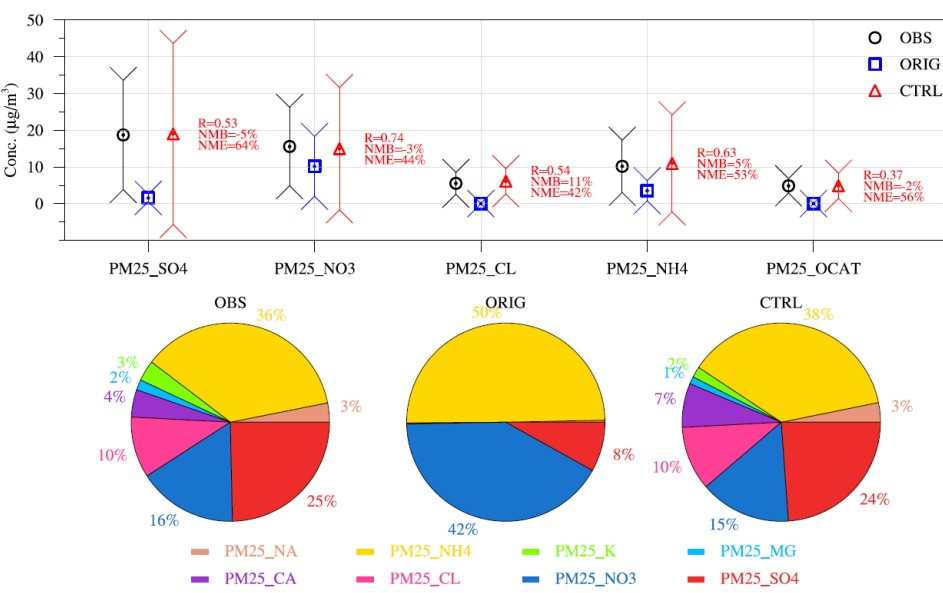

Figure 2. Significant improvement of simulations in the CTRL scenario compared with that in the ORIG scenario. Bottom panel: observed (OBS) and simulated (scenarios of ORIG and CTRL) mean electric charge fractions for fine particulate

5   sulfate (PM25_SO4, using $SO_4^{2-}$ as the surrogate), nitrate (PM25_NO3), ammonium (PM25_NH4), chloride (PM25_CL), sodium (PM25_NA), potassium (PM25_K), calcium (PM25_CA) and magnesium (PM25_MG) at TSU site during January of 2013. Top panel: observed (OBS) and simulated (scenarios of ORIG and CTRL) concentrations (both average and standard deviation are shown) for fine particulate sulfate (PM25_SO4), nitrate (PM25_NO3), chloride (PM25_CL), ammonium (PM25_NH4) and other cation components (PM25_OCAT=PM25_NA+PM25_K+PM25_CA+PM25_MG) at

10   Beijing TSU site during January of 2013.



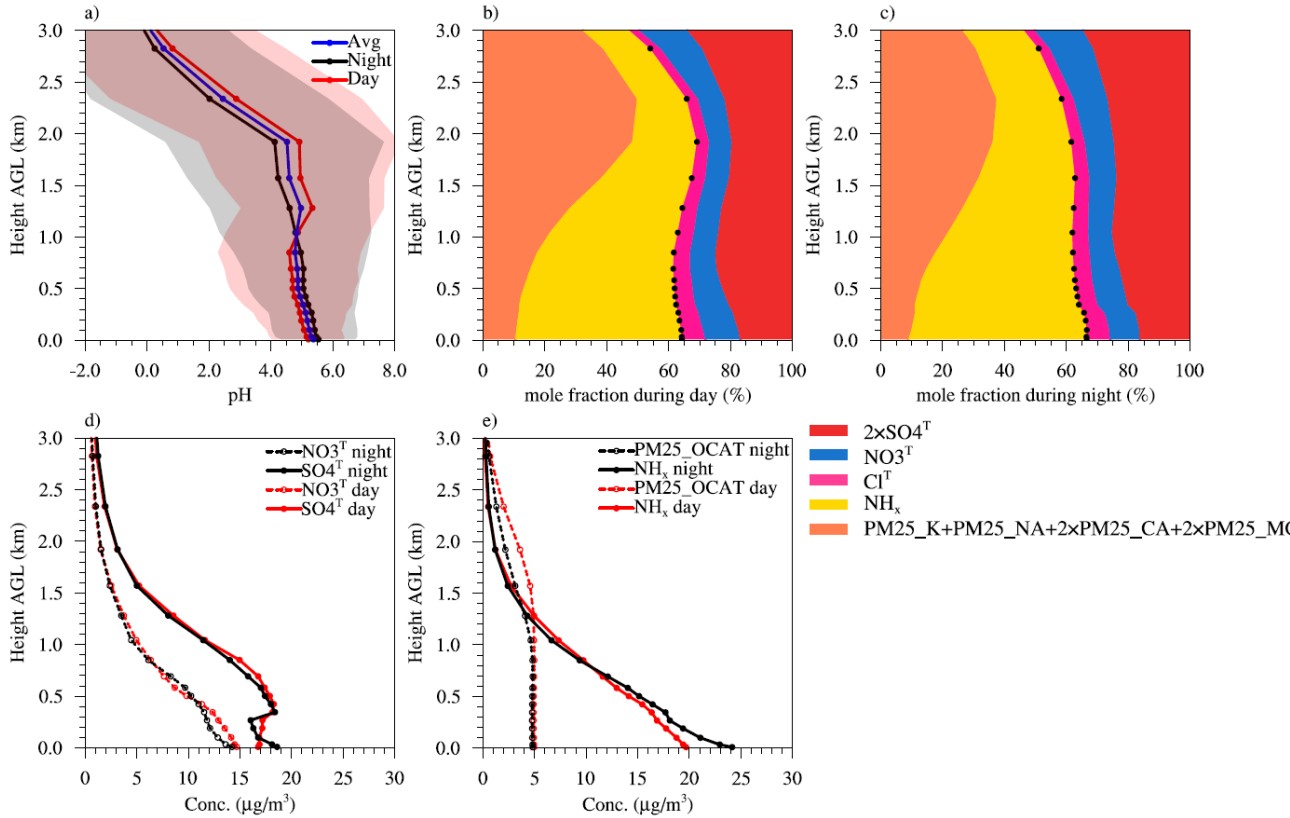

Figure 3. Vertical profiles of the fine particle pH and its potential influencing factors over Beijing TSU site. The data are simulated with the CTRL scenario during January of 2013 and presented at height above ground level (AGL). (a) Monthly average pH (blue), as well as daytime (red) and night-time (black) average pH (standard deviations are shown as light grey and light red shadings, respectively). (b) Daytime mole fractions for the electric charge of total ammonia ($NH_x$), total chloride ($Cl^T$), total sulfate ($SO4^T$), total nitrate ($NO3^T$) and different crustal components of $PM_{2.5}$, and the black dots represent the vertical profile of $cation^T/(cation^T+anion^T)$. (c) The same as in (b), but for the night-time. (d) Daytime (red) and night-time (black) concentrations of total sulfate ($SO4^T$) and total nitrate ($NO3^T$). (e) Daytime (red) and night-time (black) concentrations of $NH_x$ and PM25_OCAT.



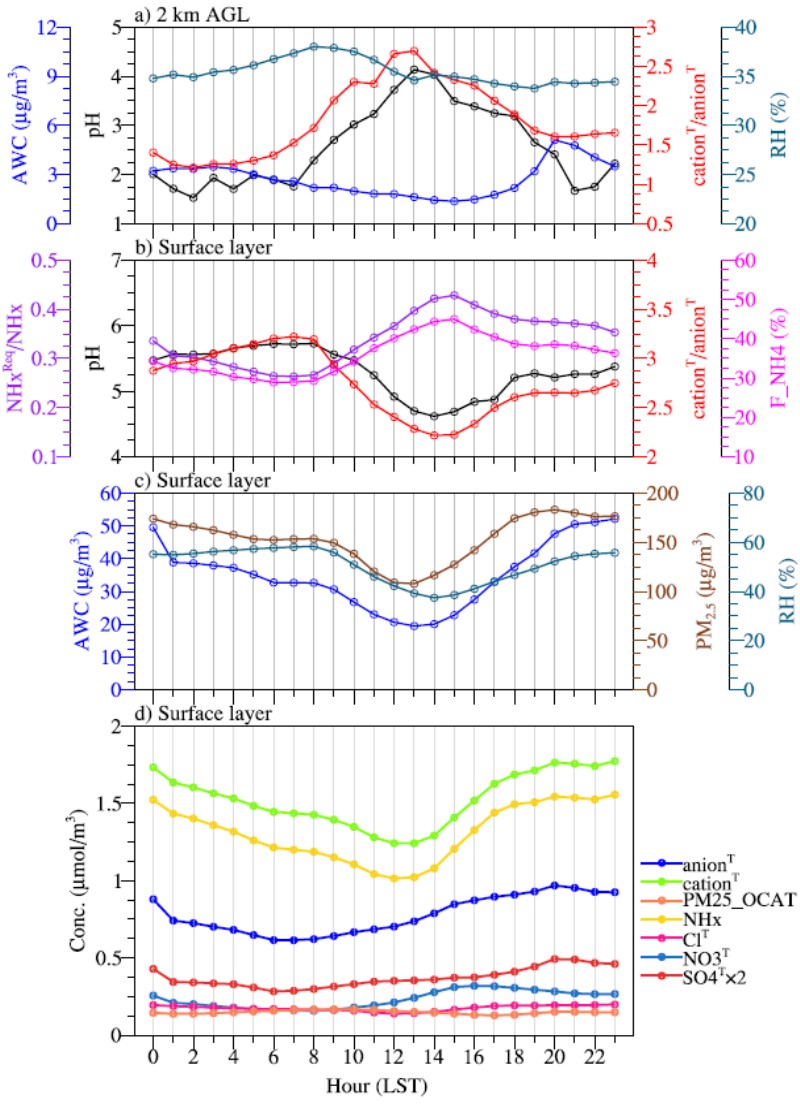

Figure 4. Monthly mean diurnal cycle for (a) fine particle pH, aerosol liquid water content (AWC), the mole ratio of total potential cation (cation$^T$) to total potential anion (anion$^T$), and relative humidity (RH) at 2 km AGL, (b) fine particle pH, the mole ratio of cation$^T$ to anion$^T$, the mole ratio of required NH$_x$ (NH$_x^{Req}$) to NH$_x$, fraction of NH$_x$ in the particle phase (F_NH4) at surface layer, (c) AWC, RH and PM$_{2.5}$ concentrations at surface layer, (d) concentrations for total sulfate (SO4$^T$), total nitrate (NO3$^T$), total chloride (Cl$^T$), total ammonia (NH$_x$), PM25_OCAT, cation$^T$ and anion$^T$ at surface layer simulated in CTRL scenario at TSU site during January of 2013.

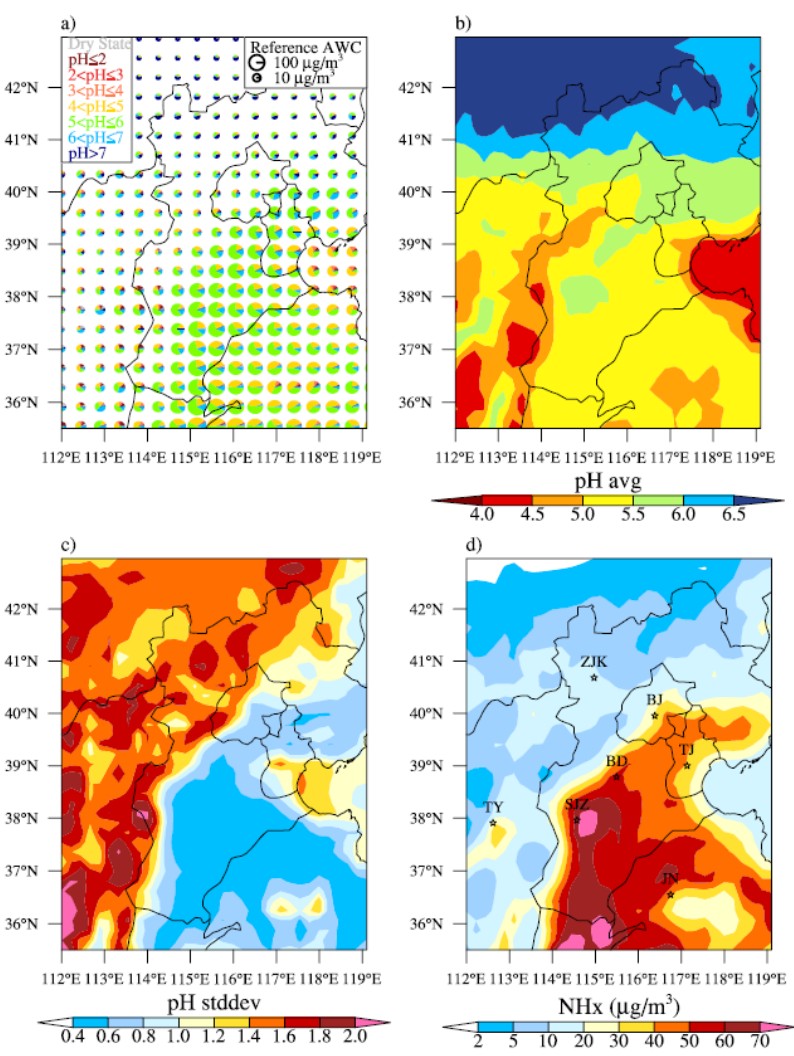

Figure 5. Fine particle pH and NH$_x$ concentrations at surface layer simulated in CTRL scenario during January of 2013: a) the frequencies for different pH ranges (the radius R for the pie chart is scaled using R/R$_{ref}$=2exp(log10(AWC/AWC$_{ref}$)), and the reference radius R$_{ref}$ and aerosol liquid water content (AWC$_{ref}$) are also shown), horizontal distribution of the average (b) as well as standard deviation (c) for pH and the concentrations for NH$_x$ (d). The locations for the 7 cities of Beijing (BJ), Tianjin (TJ), Zhangjiakou (ZJK), Baoding (BD), Shijiazhuang (SJZ), Taiyuan (TY) and Jinan (JN) are also shown.



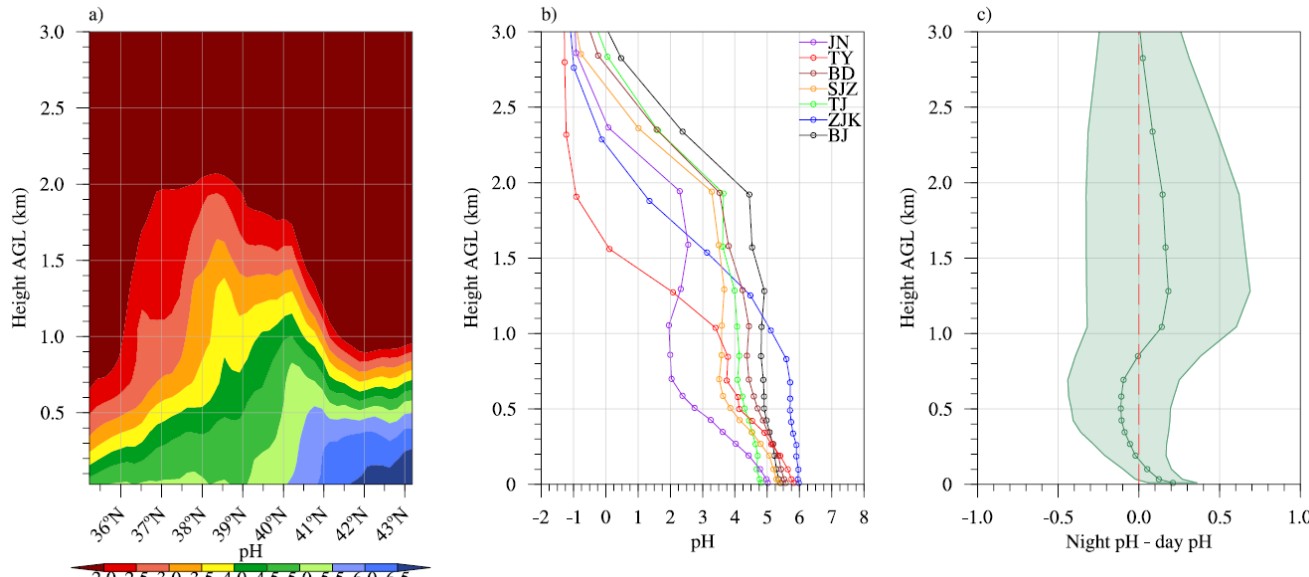

Figure 6. (a) Latitude-height cross section for the monthly mean fine particle pH averaged between 113-119°E, (b) vertical profile for the monthly mean fine particle pH over the 7 cities of Beijing (BJ), Tianjin (TJ), Zhangjiakou (ZJK), Baoding (BD), Shijiazhuang (SJZ), Taiyuan (TY) and Jinan (JN) (locations shown in Fig. 5c), (c) vertical profile for the monthly mean day-night difference of pH (standard deviation is shown as shading) for the domain-wide grid cells in CTRL scenario during January of 2013.

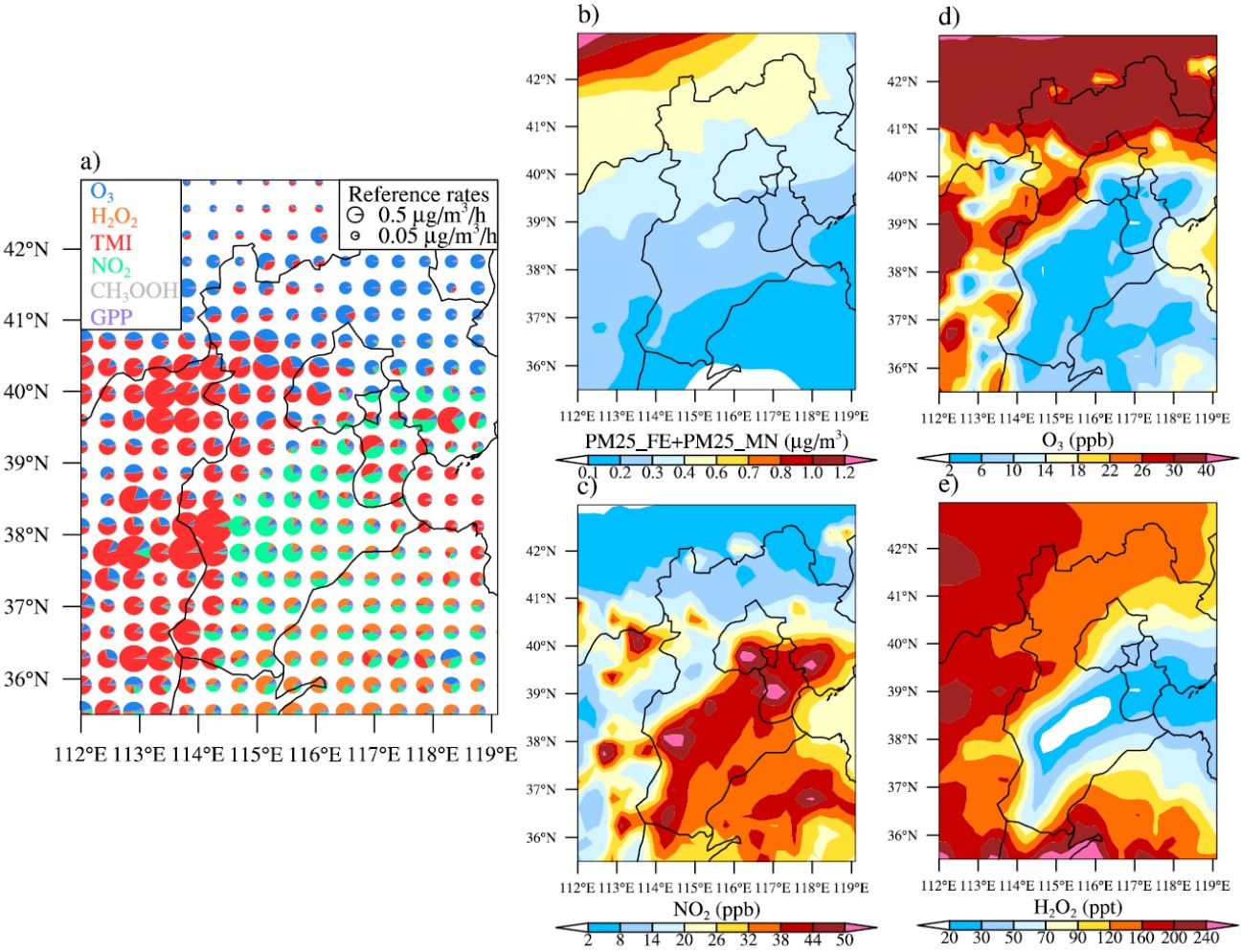

Figure 7. (a) The relative contributions of six sulfate formation pathways, namely aqueous phase oxidation by dissolved $O_3$, $H_2O_2$, TMI (in presence of $O_2$), $NO_2$ and $CH_3OOH$, as well as gas-particle partitioning of $H_2SO_4$ vapour (GPP) at surface layer simulated in CTRL scenario during January of 2013. The radius R for the pie chart in the top panel is scaled using $R/R_{ref}=2exp(log10(P/P_{ref}))$, and the reference sulfate production rate P are also shown. Horizontal distribution for the concentrations of (b) the sum of PM25_FE and PM25_MN, (c) $NO_2$, (d) $O_3$ as well as (e) $H_2O_2$ simulated in CTRL scenario during January of 2013.

Figure 8. Dominant sulfate formation pathway at surface layer (a), the vertical profile of sulfate production through different pathways (b), as well as concentrations for the oxidants of $O_3$ (c), $H_2O_2$ (d), TMI (e) and $NO_2$ (f), respectively averaged over the four dominant oxidation regimes ($O_3$, $H_2O_2$, TMI and $NO_2$, respectively, as shown in Fig. 8a) simulated in CTRL scenario during January of 2013. Contributions from different sulfate formation pathways are also shown in pie charts, and the radius R for pie chart is scaled depending on sulfate production rate P using $R/R_{ref}=2\exp(\log_{10}(P/P_{ref}))$.





Figure 9. Surface layer pH (denoted as $pH_{surf}$) and its day-night difference (denoted as $\Delta pH_{surf,night-day}$), the difference between surface layer pH and pH at ~2 km AGL (denoted as $\Delta pH_{surf-2km}$), and surface layer sulfate production rates (denoted as $P_{S(VI),surf}$) for the domain-wide grid cells south of 41°N in control experiment and sensitivity experiments during January of 2013. Contributions from different sulfate formation pathways over the domain-wide grid cells south of 41°N and the four dominant sulfate formation regimes ($O_3$, $H_2O_2$, TMI and $NO_2$, respectively, as defined in Fig. 8a) are also shown in pie charts. The radius R for pie chart is scaled depending on sulfate production rate P using $R/R_{ref}=2exp(log10(P/P_{ref}))$.



## Tables

Table 1. Rate expressions and rate constants for the S(IV) oxidation reactions in the aerosol water.

| Oxidants | Rate expression (M s⁻¹) | References |
|---|---|---|
| $O_3$ | $$r_{O_3+S(IV)} = (k_0[SO_2 \cdot H_2O] + k_1[HSO_3^-] + k_2[SO_3^{2-}])[O_3(aq)]$$ $k_0=2.4\times10^4\,M^{-1}s^{-1}$ <br> $k_1=3.7\times10^5\,M^{-1}s^{-1}$, E/R=5530 K <br> $k_2=1.5\times10^9\,M^{-1}s^{-1}$, E/R=5280 K | Seinfeld and Pandis (2016) |
| $H_2O_2$ | $$r_{H_2O_2+S(IV)} = (k_3[H^+][HSO_3^-][H_2O_2(aq)])/(1+13\cdot[H^+])$$ $k_3=7.45\times10^7\,M^{-1}s^{-1}$, E/R=4430 K | Seinfeld and Pandis (2016) |
| TMI | $$r_{TMI+S(IV)} = \begin{cases} k_4[H^+]^{-0.74}[Mn^{2+}][Fe^{3+}][S(IV)] & pH \le 4.2 \\ k_5[H^+]^{0.67}[Mn^{2+}][Fe^{3+}][S(IV)] & pH > 4.2 \end{cases}$$ $k_4=3.72\times10^7\,M^{-1.26}s^{-1}$ <br> $k_5=2.51\times10^{13}\,M^{-2.67}s^{-1}$ | Ibusuki and Takeuchi (1987) <br> Cheng et al. (2016) |
| $NO_2$[a] | $$r_{NO_2+S(IV)} = k_6[NO_2(aq)][S(IV)]$$ lower estimate: $k_{6,low}=(0.14\sim2)\times10^6\,M^{-1}s^{-1}$ <br> higher estimate: $k_{6,high}=(1.24\sim1.67)\times10^7\,M^{-1}s^{-1}$ <br> $k_6$ is the average of $k_{6,low}$ and $k_{6,high}$ | Seinfeld and Pandis (2016) <br> Lee and Schwartz (1983) <br> Clifton et al. (1988) <br> Cheng et al. (2016) |
| $CH_3OOH$ | $$r_{CH_3OOH+S(IV)} = k_7[H^+][HSO_3^-][CH_3OOH]$$ $k_7=1.75\times10^7\,M^{-2}s^{-1}$, E/R=3801 K | Walcek and Taylor (1986) |

[a] The rate coefficient $k_6$ (with a unit of $M^{-1}s^{-1}$) is expressed as:

$$k_6 = \begin{cases} (k_{6,low,1}+k_{6,high,1})/2 & pH \le 5 \\ (k_{6,low,1}+(pH-5)/0.8\times(k_{6,low,2}-k_{6,low,1})+k_{6,high,1})/2 & 5<pH \le 5.3 \\ (k_{6,low,1}+(pH-5)/0.8\times(k_{6,low,2}-k_{6,low,1})+k_{6,high,1}+(pH-5.3)/3.4\times(k_{6,high,2}-k_{6,high,1}))/2 & 5.3<pH \le 5.8 \\ (k_{6,low,2}+(pH-5.3)/3.4\times(k_{6,high,2}-k_{6,high,1}))/2 & 5.8<pH \le 8.7 \\ (k_{6,low,2}+k_{6,high,2})/2 & pH>8.7 \end{cases}$$

5    where $k_{6,low,1}=0.14\times10^6\,M^{-1}s^{-1}$, $k_{6,low,2}=2\times10^6\,M^{-1}s^{-1}$, $k_{6,high,1}=1.24\times10^7\,M^{-1}s^{-1}$, $k_{6,high,2}=1.67\times10^7\,M^{-1}s^{-1}$





Table 2. Description for different scenarios in this study.

| Number | Name | Description |
|---|---|---|
| 1 | ORIG | $emissf_{NH3}$=1, $emissf_{OCAT}$=0, $emissf_{CL}$=0, and aqueous phase oxidation in the aerosol water is excluded. |
| 2 | CTRL | Control experiment tuned to match the observations at TSU site. A fixed metastable (stable) phase state is assumed if RH is greater (not greater) than 30%. $emissf_{NH3}$=2, $emissf_{OCAT}$=4.5, $emissf_{CL}$=6, $FS_{Fe3+}$=7% and $FS_{Mn2+}$=40%. |
| 3 | MEIC_CTRL | The same as CTRL, except $emissf_{NH3}$=1 (original MEIC inventory), $FS_{Fe3+}$=0.35% and $FS_{Mn2+}$=20%. Another control experiment tuned to match the observations at TSU site. |
| 4-5 | A0, A2 | The same as CTRL, except $emissf_{NH3}$ is set to zero, halved and doubled, respectively. |
| 6-7 | OCAT0, OCAT2 | The same as CTRL, except $emissf_{OCAT}$ is set to zero, halved and doubled, respectively. |
| 8-9 | TMI0, TMI2 | The same as CTRL, except $FS_{FE3+}$ and $FS_{MN2+}$ are set to zero, halved and doubled, respectively. |
| 10-11 | CL0, CL2 | The same as CTRL, except $emissf_{CL}$ is set to zero, halved and doubled, respectively. |
| 12-13 | MSTB, STB | The same as CTRL, except a fixed metastable/stable phase state is assumed. |

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
