# Peer review of "Aerosol pH and chemical regimes of sulfate formation in aerosol water during winter haze in the North China Plain"

_Atmospheric Chemistry and Physics, 2020_

## Referee Comment (RC1) · Anonymous Referee #1 · 10 May 2020

Understanding the relative importance of sulfate formation pathways is essential for mitigation of haze pollution in China. However, there are a lot of debates on this topic. This manuscript presents a very comprehensive examination of aerosol pH and the relative importance of sulfate formation pathways during winter haze in the North China Plain. The results elucidate the dynamic changes in both pH and chemical regimes of sulfate formation. The scientific importance and presentation are of high quality, but some details need to be added before being published. The specific comments are listed below:

Page 4, line 1: in the North China Plain

[Figure]

Page 4, line 18: Sulfate is used in other places, use sulfurous instead of sulphurous to keep consistent.

Page 5, lines 5-8: This is heterogeneous uptake of NO2 on surface of fine particles, not aqueous phase chemistry. Is there any reason to include it? The purpose is not clear.

Page 7, lines 14-17: Is there any observation of dissolved FE3+ or MN2+ to adjust FSFE3+ and FSMN2+. In Wang et al. (2016), observations of Fe and Mn are provided. It is not very convincing to adjust FSFE3+ and FSMN2+ based on sulfate observation here.

https://www.pnas.org/content/113/48/13630

---

## Referee Comment (RC2) · Anonymous Referee #2 · 13 Jun 2020

General Comments:

This study implemented a new aerosol water chemistry module (AWAC) in the WRF-Chem model, and aimed to understand the mechanisms of haze formation over China, in particular, to examine the relative roles of multiphase chemical reactions in aerosol water on particulate sulfate production, which is mainly related to the questions about aerosol pH. They investigated the spatial and temporal distributions of pH around Beijing with the model, and found that the rapid production of sulfate in the NCP can be maintained with the pH range of 4.2-5.7. This is a very interesting and important work. Scientifically, it is still under debate. The analysis of modeling results provided some

evidence. However, I still have some questions about the uncertainty of results and the robustness of conclusion. More analysis and clarifications are needed before publication.

Specific Comments:

1. As the authors also agreed, the pH may be one of the key factors controlling the AWAC processes. However, unfortunately, there is no direct measurement of pH for evaluation. Currently, most studies used the model to calculate the pH, which makes the pH estimation dependent on modules. It is good to couple ISORROPIA II into WRF-Chem, but we still cannot rule out the dependence of pH calculation on this module. In WRF-Chem, the existing module for pH calculation is MOSAIC. Did the authors estimate the pH with MOSAIC and compare the values with ISORROPIA? Are they consistent?

2. For evaluation, since NH3 and NH4- are so important in this AWAC system, could authors evaluate both of them? In Fig. 1, I didn't find the evaluation of NH4- and NH3. In Fig. 2, for PM25_OCAT, why not evaluate the absolute values of each components such as K, MG, CA? The emission factor of OCAT is multiplied by 4.5 to match observation. How is this applied? Do you apply it to the total dust emission? This is a huge factor. Did you evaluate the dust mass/AOD over the dust source region to confirm this?

3. In Fig. 1 and 2, although the added AWAC significantly increased sulfate production and the mean is closer to the observation, however, it is evident that the model still missed many events. This reflects that there are still some other important processes/mechanisms are missed in the model. Therefore, is it reasonable to use the observation to constrain the model AWAC process? i.e., there may be other processes contributing to the sulfate mass concentration more than AWAC? Please add some explanation and discussion. Related question, any evidence of significant contribution of AWAC on sulfate production in other events in recent year (2017, 2018, 2019)?

4. Line 21 of page 3, "except for" to "besides"?

5. Table 2, the description of scenarios includes "halved". It seems to me that there are only two cases: zero and doubled.

---

## Author Response (AR1)

**Response to Anonymous Referee #1**

Understanding the relative importance of sulfate formation pathways is essential for mitigation of haze pollution in China. However, there are a lot of debates on this topic. This manuscript presents a very comprehensive examination of aerosol pH and the relative importance of sulfate formation pathways during winter haze in the North China Plain. The results elucidate the dynamic changes in both pH and chemical regimes of sulfate formation. The scientific importance and presentation are of high quality, but some details need to be added before being published. The specific comments are listed below:

We thank the reviewer for the very valuable and constructive comments, which help us to improve the study and manuscript. Please find our point-by-point response (black) and the corresponding revisions (blue and *Italic*) below.

1.  Page 4, line 1: in the North China Plain

**Response:**

Thanks. We have revised it accordingly:

*"regimes for sulfate formation may indeed co-exist in the North China Plain"*

2.  Page 4, line 18: Sulfate is used in other places, use sulfurous instead of sulphurous to keep consistent.

**Response:**

Thanks. We have replaced the "sulphurous" by the "sulfurous" throughout the manuscript.

3.  Page 5, lines 5-8: This is heterogeneous uptake of $NO_2$ on surface of fine particles, not aqueous phase chemistry. Is there any reason to include it? The purpose is not clear.

**Response:**

Thanks for pointing out this issue. In this study, we focus on the aerosol pH as well as detailed mechanisms for the sulfate formation (the relevant reactions rates highly depend on aerosol pH) in aerosol water. In order to better predict aerosol pH, it is necessary to reproduce the observed aerosol loadings for sulfate, nitrate and other components. And we have tested that if the heterogeneous formation of nitrate was ignored, nitrate concentrations would be underestimated. Considering that the explicit mechanisms for the aqueous phase production of nitrate are very complicated (Herrmann et al., 1999;Herrmann et al., 2005), and beyond the scope of our current study, we have adopted a parameterization scheme to simulate the heterogeneous formation of nitrate (Zheng et al., 2015;Chen et al., 2016). We have clarified this issue in the third paragraph of Section 2.2 in the revised manuscript:

*"The CTRL scenario is expected to reproduce the observed fine particle compositions (including sulfate, nitrate, ammonium, chloride and crustal components) and gas phase pollutants, and thus more reliably predict the spatio-temporal distribution of pH, AWC and sulfate production."*

4.  Page 7, lines 14-17: Is there any observation of dissolved FE3+ or MN2+ to adjust FSFE3+ and FSMN2+. In Wang et al. (2016), observations of Fe and Mn are provided. It is not very convincing to adjust FSFE3+ and FSMN2+ based on sulfate observation here.
https://www.pnas.org/content/113/48/13630

**Response:**

Very good suggestion, thanks. Soluble Fe/Mn concentrations are often measured and reported in studies focusing on dust events (e.g., Takahashi et al., 2011;Shi et al., 2012;Schroth et al., 2009;Ravelo-Perez et

al., 2016). However, during the urban haze episodes, the concentrations as well as sources for Fe/Mn ions in aerosol water remain not well constrained and understood, although the observed concentrations of total Fe/Mn elements in urban $PM_{2.5}$ have been reported in some previous studies (e.g., Sun et al., 2006;Wang et al., 2006;Chen et al., 2017). Validation of the simulated $Fe^{3+}/Mn^{2+}$ ions in aerosol water is indeed limited by the lack of observations in our study. Nonetheless, we have compared the simulated soluble Fe/Mn concentrations with the observed data reported in Wang et al. (2016). As shown in Table R1, the simulated concentrations for $Fe^{3+}/Mn^{2+}$ at the Beijing TSU site and the Yan'an city are at the same or similar order of magnitude as the observations at a Xi'an urban site (note that the Xi'an site is outside of our simulated domain, and inside the simulated domain, Yan'an city is the nearest city to Xi'an).

Table R1. Comparison of the soluble Fe/Mn concentrations in aerosol water

|  | This study | This study | Wang et al. (2016) |
|---|---|---|---|
| Time | January 2013 | January 2013 | November-December 2012 |
| Site location | Beijing | Yan'an | Xi'an |
| Soluble Fe (ng/m³) | / | / | 1.5-16 |
| Soluble Mn (ng/m³) | / | / | 10-41 |
| $Fe^{3+}$ (ng/m³) | 3.2 | 0.6 | / |
| $Mn^{2+}$ (ng/m³) | 3.6 | 3.3 | / |

In this study, we have adopted 7% and 40% for $FS_{FE3+}$ and $FS_{MN2+}$, respectively. The observed values for $FS_{FE}$ and $FS_{MN}$ both show a large variability (1-10% and 20-60%, respectively), depending on dust mineralogy and atmospheric aging (Journet et al., 2008;Schroth et al., 2009;Johnson et al., 2010;Shi et al., 2012;Claquin et al., 1999;Meskhidze et al., 2003;Takahashi et al., 2011;Duvall et al., 2008;Baker et al., 2006;Hsu et al., 2010). Our adopted values are within these ranges. We have also investigated the potential changes in predicted aerosol pH and sulfate production relevant with the uncertainties in TMI concentrations. Sensitivity tests indicate that sulfate formation is rather sensitive to the availability of TMI species, and TMI concentrations need to be better constrained in future observational and modeling studies. We have further clarified this issue in the third paragraph of the Discussion Section:

*"Our results indicate that sulfate production is rather sensitive to the availability of TMI species. Unfortunately, the concentrations as well as sources for TMI species in aerosol water during haze episodes remain not well constrained and understood. The simulated mean concentration for $Fe^{3+}$ and $Mn^{2+}$ in $PM_{2.5}$ at the Beijing TSU site is 3.2 and 3.6 ng/m³, respectively, and are smaller than the observed concentrations for soluble Fe and Mn (1.5-16 and 10-41 ng/m³, respectively) in $PM_{2.5}$ at a Xi'an site (Wang et al., 2016). Note that $Fe^{3+}/Mn^{2+}$ ions also have an anthropogenic source, and were estimated to account for 10-30% in Beijing (Shao et al., 2019). Furthermore, the soluble Fe/Mn speciation (including $Fe^{3+}-Fe^{2+}$, $Mn^{2+}-Mn^{3+}-Mn^{4+}$ cycling) depends on dust mineralogy, particle acidity and heterogeneous redox reactions (Takahashi et al., 2011;Schroth et al., 2009), and is very difficult to be explicitly treated. Also the activity coefficients for $Fe^{3+}/Mn^{2+}$ ions under the high ionic strength environment might differ (Cheng et al., 2016). The treatment of TMI pathway should be further improved in future studies."*

**References**

Baker, A. R., Jickells, T. D., Witt, M., and Linge, K. L.: Trends in the solubility of iron, aluminium,

manganese and phosphorus in aerosol collected over the Atlantic Ocean, Marine Chemistry, 98, 43-58, 10.1016/j.marchem.2005.06.004, 2006.

Chen, D., Liu, Z., Fast, J., and Ban, J.: Simulations of sulfate–nitrate–ammonium (SNA) aerosols during the extreme haze events over northern China in October 2014, Atmos. Chem. Phys., 16, 10707-10724, 2016.

Chen, F., Zhang, X., Zhu, X., Zhang, H., Gao, J., and Hopke, P. K.: Chemical Characteristics of PM 2.5 during a 2016 Winter Haze Episode in Shijiazhuang, China, Aerosol and Air Quality Research, 17, 368-380, 2017.

Claquin, T., Schulz, M., and Balkanski, Y. J.: Modeling the mineralogy of atmospheric dust sources, J Geophys Res-Atmos, 104, 22243-22256, Doi 10.1029/1999jd900416, 1999.

Duvall, R. M., Majestic, B. J., Shafer, M. M., Chuang, P. Y., Simoneit, B. R. T., and Schauer, J. J.: The water-soluble fraction of carbon, sulfur, and crustal elements in Asian aerosols and Asian soils, Atmos. Environ., 42, 5872-5884, 10.1016/j.atmosenv.2008.03.028, 2008.

Herrmann, H., Ervens, B., Nowacki, P., Wolke, R., and Zellner, R.: A chemical aqueous phase radical mechanism for tropospheric chemistry, Chemosphere, 38, 1223-1232, 1999.

Herrmann, H., Tilgner, A., Barzaghi, P., Majdik, Z., Gligorovski, S., Poulain, L., and Monod, A.: Towards a more detailed description of tropospheric aqueous phase organic chemistry: CAPRAM 3.0, Atmospheric Environment, 39, 4351-4363, 2005.

Hsu, S. C., Wong, G. T. F., Gong, G. C., Shiah, F. K., Huang, Y. T., Kao, S. J., Tsai, F. J., Lung, S. C. C., Lin, F. J., Lin, I. I., Hung, C. C., and Tseng, C. M.: Sources, solubility, and dry deposition of aerosol trace elements over the East China Sea, Marine Chemistry, 120, 116-127, 10.1016/j.marchem.2008.10.003, 2010.

Johnson, M. S., Meskhidze, N., Solmon, F., Gasso, S., Chuang, P. Y., Gaiero, D. M., Yantosca, R. M., Wu, S. L., Wang, Y. X., and Carouge, C.: Modeling dust and soluble iron deposition to the South Atlantic Ocean, J Geophys Res-Atmos, 115, Artn D15202
10.1029/2009jd013311, 2010.

Journet, E., Desboeufs, K. V., Caquineau, S., and Colin, J. L.: Mineralogy as a critical factor of dust iron solubility, Geophysical Research Letters, 35, Artn L07805
10.1029/2007gl031589, 2008.

Meskhidze, N., Chameides, W., Nenes, A., and Chen, G.: Iron mobilization in mineral dust: Can anthropogenic SO2 emissions affect ocean productivity?, Geophysical Research Letters, 30, 2003.

Ravelo-Perez, L. M., Rodriguez, S., Galindo, L., Garcia, M. I., Alastuey, A., and Lopez-Solano, J.: Soluble iron dust export in the high altitude Saharan Air Layer, Atmos. Environ., 133, 49-59, 10.1016/j.atmosenv.2016.03.030, 2016.

Schroth, A. W., Crusius, J., Sholkovitz, E. R., and Bostick, B. C.: Iron solubility driven by speciation in dust sources to the ocean, Nature Geoscience, 2, 337-340, 10.1038/Ngeo501, 2009.

Shi, Z. B., Krom, M. D., Jickells, T. D., Bonneville, S., Carslaw, K. S., Mihalopoulos, N., Baker, A. R., and Benning, L. G.: Impacts on iron solubility in the mineral dust by processes in the source region and the atmosphere: A review, Aeolian Research, 5, 21-42, 10.1016/j.aeolia.2012.03.001, 2012.

Sun, Y., Zhuang, G., Tang, A. A., Wang, Y., and An, Z.: Chemical characteristics of PM2.5 and PM10 in haze-fog episodes in Beijing, Environ Sci Technol, 40, 3148-3155, 10.1021/es051533g, 2006.

Takahashi, Y., Higashi, M., Furukawa, T., and Mitsunobu, S.: Change of iron species and iron solubility in Asian dust during the long-range transport from western China to Japan, Atmos. Chem. Phys., 11, 11237-11252, 10.5194/acp-11-11237-2011, 2011.

Wang, G., Zhang, R., Gomez, M. E., Yang, L., Levy Zamora, M., Hu, M., Lin, Y., Peng, J., Guo, S., Meng, J., Li, J., Cheng, C., Hu, T., Ren, Y., Wang, Y., Gao, J., Cao, J., An, Z., Zhou, W., Li, G., Wang, J., Tian, P., Marrero-Ortiz, W., Secrest, J., Du, Z., Zheng, J., Shang, D., Zeng, L., Shao, M., Wang, W., Huang, Y., Wang, Y., Zhu, Y., Li, Y., Hu, J., Pan, B., Cai, L., Cheng, Y., Ji, Y., Zhang, F., Rosenfeld, D., Liss, P. S., Duce, R. A., Kolb, C. E., and Molina, M. J.: Persistent sulfate formation from London Fog to Chinese haze, Proc Natl Acad Sci U S A, 113, 13630-13635, 10.1073/pnas.1616540113, 2016.

Wang, Y., Zhuang, G. S., Sun, Y. L., and An, Z. S.: The variation of characteristics and formation mechanisms of aerosols in dust, haze, and clear days in Beijing, Atmos. Environ., 40, 6579-6591, 10.1016/j.atmosenv.2006.05.066, 2006.

Zheng, B., Zhang, Q., Zhang, Y., He, K. B., Wang, K., Zheng, G. J., Duan, F. K., Ma, Y. L., and Kimoto, T.: Heterogeneous chemistry: a mechanism missing in current models to explain secondary inorganic aerosol formation during the January 2013 haze episode in North China, Atmos. Chem. Phys., 15, 2031-2049, 10.5194/acp-15-2031-2015, 2015.

**Response to Anonymous Referee #2**

This study implemented a new aerosol water chemistry module (AWAC) in the WRF-Chem model, and aimed to understand the mechanisms of haze formation over China, in particular, to examine the relative roles of multiphase chemical reactions in aerosol water on particulate sulfate production, which is mainly related to the questions about aerosol pH. They investigated the spatial and temporal distributions of pH around Beijing with the model, and found that the rapid production of sulfate in the NCP can be maintained with the pH range of 4.2-5.7. This is a very interesting and important work. Scientifically, it is still under debate. The analysis of modeling results provided some evidence. However, I still have some questions about the uncertainty of results and the robustness of conclusion. More analysis and clarifications are needed before publication.

We thank the reviewer for the very valuable and constructive comments, which help us to improve the study and manuscript. Please find our point-by-point response (black) and the corresponding revisions (blue and *Italic*) below.

**Specific Comments:**

1. As the authors also agreed, the pH may be one of the key factors controlling the AWAC processes. However, unfortunately, there is no direct measurement of pH for evaluation. Currently, most studies used the model to calculate the pH, which makes the pH estimation dependent on modules. It is good to couple ISORROPIA II into WRF-Chem, but we still cannot rule out the dependence of pH calculation on this module. In WRF-Chem, the existing module for pH calculation is MOSAIC. Did the authors estimate the pH with MOSAIC and compare the values with ISORROPIA? Are they consistent?

**Response:**

Good question, thanks. In the original WRF-Chem model Version 3.8, both MADE/SORGAM/ISORROPIA aerosol scheme and MOSAIC aerosol scheme could be used to simulate the aerosol thermodynamics (including aerosol pH and water content). Previous field campaign studies have used the ISORROPIA II model to simulate the aerosol thermodynamics during the haze or non-haze periods in the North China Plain, and reported reasonable model performance (Song et al., 2018;Shi et al., 2017;Liu et al., 2017;Ding et al., 2019;Guo et al., 2017). However, the applicability of MOSAIC aerosol scheme specifically in simulating the aerosol thermodynamics in the North China Plain remains rarely reported. Thus in this study, we have chosen the MADE/SORGAM/ISORROPIA aerosol scheme, and further updated the default ISORROPIA model (Nenes et al., 1998) with the improved version (Fountoukis and Nenes, 2007;Song et al., 2018). The simulated mean pH for different scenarios in our study ranges between 4.2 and 5.7 in the North China Plain, and is comparable with the mean pH values reported in previous relevant studies (e.g., Liu et al., 2017;Shi et al., 2017;Song et al., 2018;Ding et al., 2019).

In the MOSAIC framework (Zaveri et al., 2008), the Multicomponent Equilibrium Solver for Aerosols (MESA) is used to simulate the aerosol thermodynamics (Zaveri et al., 2005a;Zaveri et al., 2005b). The algorithm of MESA model differs significantly from that of ISORROPIA II, including the chemical species involved (Potassium and Magnesium is excluded in MESA), determination of activity coefficient and mutual deliquescence relative humidity (MDRH), and the treatment of phase state and Kelvin Effect. Pye et al. (2020) has compared the pH values estimated by the box-model version of MOSAIC and ISORROPIA II (with the identical modeling input), and they found that the average aerosol pH differed

by 0.3 unit (the difference in pH could be up to 1 unit, and was greater with the decreasing relative humidity). It seems that there is no significant disparity in terms of predicting pH between models of MOSAIC and ISORROPIA. And we agree with the reviewer that it is very important and interesting to compare and analyze the results of different aerosol schemes during the severe haze episodes simulated in our study. But such issue is beyond the research scope of our current study, and it requires considerable efforts to couple the aerosol water chemistry module (AWAC) with the MOSAIC aerosol scheme. Nonetheless, we have added some relevant discussion as a caveat of our study in the Conclusion Section of the revised manuscript:

*"Uncertainties relevant with the algorithms to solve the aerosol thermodynamics, including the treatment of non-ideality, size effects, phase state, mixing state, the interactions between inorganic compounds and organic compounds, as well as phase separation, should also be addressed in future studies."*

2. For evaluation, since NH3 and NH4+ are so important in this AWAC system, could authors evaluate both of them? In Fig. 1, I didn't find the evaluation of NH4+ and NH3.

**Response:**

Good suggestion, thanks. Unfortunately, the observational data for ammonia concentrations in the North China Plain during January 2013 is unavailable. Nonetheless we have compared the simulated ammonia concentrations against the observations at other urban Beijing sites during the wintertime in other years:

Table R1. Modelled and observed $NH_3$, total $NH_x$ ($TNH_x$) and fraction of $NH_x$ in the particle phase (F_NH4) at urban Beijing sites [a].

| | $NH_3$ mean (ppb) | $NH_3$ median (ppb) | $TNH_x$ mean (ppb) | $TNH_x$ median (ppb) | F_NH4 mean (%) | F_NH4 median (%) |
|---|---|---|---|---|---|---|
| MEIC_CTRL [b] | 4.9 | 5.0 | 17.5 | 12.0 | 72 | 70 |
| CTRL [b] | 15.5 | 13.5 | 28.3 | 21.2 | 45 | 61 |
| Meng et al. (2011) [c] | 10.3 | / | / | / | / | / |
| Liu et al. (2017) [d] | 22.0 | / | / | / | / | / |
| Song et al. (2018) [e] | / | 18.0 | / | 39.1 | / | 54 |

[a] The modelling and measuring time differs, including months of November, December, January and February. Nonetheless, estimated emissions and observed concentrations of $NH_3$ in one study (e.g., Meng et al., 2011;Zhang et al., 2018) both have a minor difference among these months.
[b] Monthly mean value at Tsinghua University site (referred to Beijing site) during January of 2013.
[c] Mean value at Chinese Academy of Meteorological Sciences site during wintertime from 2008 to 2009.
[d] Mean value at Peking University site during November and December in both 2015 and 2016.
[e] Median value at Institute of Atmospheric Physics site from November to December of 2014.

As shown in Table R1, doubling the $NH_3$ emissions better match the observed ammonia and ammonium concentrations. Furthermore, Kong et al. (2019) estimated that the MEIC inventory (used as the anthropogenic emission inventory in our study) under-predicted $NH_3$ emissions by about 40% in the North China Plain. Thus, doubling the $NH_3$ emissions seems a reasonable assumption. We keep Table R1 in the Supplement, and further clarify this issue in the Section 2.2 of the revised manuscript:

*"As shown in Table S7 of Supplement, compared with the scenario using the default MEIC emission data, the CTRL scenario (with doubled $NH_3$ emissions) better matches both the observed ammonia and ammonium concentrations at urban Beijing sites during wintertime (Meng et al., 2011;Liu et al., 2017;Song et al., 2018)."*

In Fig. 2, for PM25_OCAT, why not evaluate the absolute values of each components such as K, MG, CA? The emission factor of OCAT is multiplied by 4.5 to match observation. How is this applied? Do you apply it to the total dust emission? This is a huge factor. Did you evaluate the dust mass/AOD over the dust source region to confirm this?

**Response:**

We thank the reviewer for raising this concern and the very good comment. The dust scheme we used is the improved GOCART dust scheme which is coupled with the MADE/SORGAM aerosol scheme. This dust scheme has been described and evaluated in Zhao et al. (2010) and Zhao et al. (2013). We do not change any of this dust scheme (i.e., the simulated total dust flux is unchanged). WRF/Chem prescribes a specific mass ratio of the fine mode dust emission to the total dust emission, and we further adopt the speciation fractions for the fine mode crustal particles of K, Na, Ca and Mg within the East Asian fine mode dust from Dong et al. (2016). However, Dong et al. (2016) indicated that observed fine particles have a considerably higher mass contribution within the East Asian dust than the chemical transport model (CTM) prescribes. With sensitivity tests, we found that multiply the fine mode dust speciation fractions for K, Na, Ca and Mg by merely one factor of 4.5 could better match the observed PM25_OCAT concentrations at Beijing TSU site.

We have clarified how to tune the PM25_OCAT concentrations in the second and third paragraph of Section 2.2 in the revised manuscript:

*"WRF/Chem prescribes a specific mass ratio of the fine mode dust emission to the total dust emission, and we further adopt the fine mode dust emission speciation profiles from Dong et al. (2016). The mass fraction for fine particle components of K, Na, Ca, Mg, Fe and Mn minerals (denoted as PM25_K, PM25_NA, PM25_CA, PM25_MG, PM25_FE and PM25_MN, respectively) from dust source are set as 3.77%, 3.94%, 7.94%, 0.80%, 2.43%, and 0.063%, respectively."*

*"To match the observations of PM25_OCAT, the $emissf_{OCAT}$ is set to 4.5 (the total dust emission is unchanged), and Dong et al. (2016) indicated that observed fine particles should have a considerably higher mass contribution within the East Asian dust than the chemical transport model prescribes."*

Nonetheless, we have compared the simulated $AOD_{550nm}$ with the MODIS as well as AERONET $AOD_{550nm}$ data during January 2013. As shown in Figure R1, the MODIS AOD data is always missing in the vicinity of dust source regions (especially the Gobi Desert), while WRF/Chem simulated an AOD of ~0.2-0.4 there. The model could well reproduce the observed pattern in day-to-day changes of AOD downwind to the Western Pacific. But during the severely polluted episodes, the simulated AOD is not overestimated but rather underestimated especially over the Beijing-Tianjin-Hebei area. Moreover, the simulated AOD at the Dalanzadgad (the capital of SouthGobi Aimag in Mongolia) site is 0.027±0.007, and is also lower than the AERONET AOD data ([https://aeronet.gsfc.nasa.gov/cgi-bin/draw_map_display_aod_v3](https://aeronet.gsfc.nasa.gov/cgi-bin/draw_map_display_aod_v3)) of 0.076. Results show that multiplying the fine mode dust speciation fractions for K, Na, Ca and Mg by a factor of 4.5 does not lead to the systematic overestimation of AOD. Even though large uncertainties exist for the simulation of dust events, the results of sensitivity tests in our study show that both the diurnal cycle pattern and vertical profile pattern for pH are consistent with varying dust emissions, meanwhile the rapid production of sulfate could be maintained.

[Figure]

Figure R1. Comparison of MODIS AOD and simulated AOD during January of 2013.

Figure R2 compares the observed PM25_K, PM25_NA, PM25_MG, and PM25_CA with the simulated results (OBS vs. CTRL). The model in general reasonably predicts these individual crustal species, but with a slight overestimation of PM25_CA. We have conducted an additional control simulation tuned to match the observed PM25_K, PM25_MG, PM25_NA and PM25_CA at Beijing TSU site (OCAT_CTRL in Figure R2). The average pH at the surface in OCAT_CTRL case (5.0±0.6) is only about 0.2 unit lower than the CTRL case (5.2±0.5).

[Figure]

Figure R2. Observed (OBS) and simulated (scenarios of CTRL and OCAT_CTRL) mean electric charge fractions for fine particulate sulfate (PM25_SO4, using $SO_4^{2-}$ as the surrogate), nitrate (PM25_NO3), ammonium (PM25_NH4), chloride (PM25_CL), sodium (PM25_NA), potassium (PM25_K), calcium (PM25_CA) and magnesium (PM25_MG) at TSU site during January of 2013.

3. In Fig. 1 and 2, although the added AWAC significantly increased sulfate production and the mean is closer to the observation, however, it is evident that the model still missed many events. This reflects that there are still some other important processes/mechanisms are missed in the model. Therefore, is it reasonable to use the observation to constrain the model AWAC process? i.e., there may be other processes contributing to the sulfate mass concentration more than AWAC? Please add some explanation and discussion.

**Response:**

Very good question, thanks. The importance of aerosol water phase production of sulfate has been widely accepted (Li et al., 2017;Chen et al., 2016;Zheng et al., 2015;Cheng et al., 2016;Zhang et al., 2015;Wang et al., 2016;Shao et al., 2019;Xue et al., 2019;Gen et al., 2019;Chen et al., 2019;Wu et al., 2019), and we think that during the severe haze episodes, implementing the heterogeneous reactions in aerosol water might be a key to reduce the gap between modelled sulfate concentrations and observations. After implementing the AWAC, the model performance is significantly improved, and the NMB between observed sulfate-nitrate-ammonium is reduced from -40%~-90% to $\pm5\%$. However, as pointed by the reviewer, discrepancies still remain in some events, which may be due to the uncertainties in the treatment of emission, transport (i.e., advection and turbulent mixing), removal (dry and wet deposition), and also the other potential sulfate formation pathways.

Our study focuses on investigating the characterises in the spatio-temporal distribution for aerosol pH as well as sulfate formation budget, and also the uncertainties relevant with assumptions for input parameters. In the CTRL scenario, we tune the input parameters relevant with concentrations for sulfate and other fine particle components (nitrate, ammonium, chlorides and crustal species) to better constrain the spatio-temporal distribution of aerosol pH. The different sources of uncertainties have been tested (Section 3.6). Specifically, we have used the observed sulfate to constrain the TMI formation pathway. Here, we have added one extra simulation of TMI0.5 (both $FS_{FE3+}$ and $FS_{MN2+}$ are halved) to further investigate the uncertainty relevant with TMI concentrations. As shown in Figure 9 of the revised manuscript, our conclusion in the manuscript regarding the diurnal cycle pattern and vertical profile pattern for aerosol pH, as well as the co-existence of multiple sulfate regime and how they interact with pH is consistent in the scenarios of TMI0, TMI0.5 and TMI2.

We have further clarified this issue in the third paragraph of Section 2.2 in the revised manuscript:

*"The CTRL scenario is expected to reproduce the observed fine particle compositions (including sulfate, nitrate, ammonium, chloride and crustal components) and gas phase pollutants, and thus more reliably predict the spatio-temporal distribution of pH, AWC and sulfate production … Note that the assumption behind tuning only $FS_{FE3+}$ and $FS_{MN2+}$ to better agree with observed sulfates, is that the model could reasonably simulate the concentrations for other oxidants (e.g., OH, $H_2O_2$, $O_3$ and $NO_2$), thus the deviation from observation can be attributed to the uncertainties in representation of TMI pathway. Note that uncertainties in the emission, transport (i.e., advection and turbulent mixing), removal (dry and wet deposition) and sulfate formation in other phases could also contribute to the discrepancies between modeling results and observations. Nonetheless, this study does not aim at estimating the exact values for aerosol pH and sulfate formation budget. Instead, this study focuses on the characterises in the spatio-temporal distribution for aerosol pH as well as sulfate formation budget, and also the uncertainties relevant with assumptions for input parameters."*

Related question, any evidence of significant contribution of AWAC on sulfate production in other events in recent year (2017, 2018, 2019)?

**Response:**

A very good and interesting question, thanks. The importance of aerosol water phase production of sulfate has been widely discussed and accepted (e.g., Li et al., 2017;Chen et al., 2016;Zheng et al., 2015;Cheng et al., 2016;Wang et al., 2016;Shao et al., 2019;Wu et al., 2019) during the severe haze episodes from early to middle 2010s (Table R2).

Table R2. Selection of the studies focusing on the heterogeneous reactions during the severe haze episodes from early to middle 2010s in China.

| Reference | Study time period and area |
|---|---|
| Zheng et al. (2015) | January 2013 in Beijing–Tianjin–Hebei area |
| Chen et al. (2016) | October 2014 in North China Plain |
| Cheng et al. (2016) | January 2013 in Beijing |
| Wang et al. (2016) | 17 November to 12 December of 2012 in Xi'an |
| | 21 January to 4 February of 2015 in Beijing |
| Li et al. (2017) | 16 to 27 December 2013 in the Guanzhong basin |
| | 13 to 21 January 2014 in Beijing–Tianjin–Hebei area |
| Shao et al. (2019) | 18 October 2014 to 17 January 2015 in the whole China |
| Wu et al. (2019) | Wintertime of 2015 in the North China Plain |

However, the contribution of heterogeneous reactions to sulfate formation in recent years (2017-2020) remains rarely studied and quantified. Note that the air pollution over China has been remarkably mitigated in recent years since the implementation of Clean Air Action in 2013 (Fan et al., 2020;Zhang et al., 2019;Wang et al., 2019;Hou et al., 2019;Cheng et al., 2019). And Zheng et al. (2018) estimated that during 2013–2017, China's anthropogenic emissions decreased by ~60%, ~20% and ~35% for sulfur dioxide ($SO_2$), nitrogen oxides ($NO_x$) and $PM_{2.5}$, respectively. Unlike the negative feedback between aerosol loadings and their photochemical production (Kong et al., 2015), the multiphase reactions induce a positive feedback mechanism, i.e., higher particle matter levels lead to more aerosol water, which accelerates sulfate production and further increases the aerosol concentration (Cheng et al., 2016). The role of heterogeneous reactions might exhibit a weakening trend for the inter-annual variation with the decreasing emissions.

4. Line 21 of page 3, "except for" to "besides"?
**Response:**
Thanks for pointing out this typo, and we have corrected it.

5. Table 2, the description of scenarios includes "halved". It seems to me that there are only two cases: zero and doubled.
**Response:**
Thanks for pointing out this typo, and we have corrected it.

20    provides the concentrations of gaseous precursors (e.g., $SO_2$, $NO_2$ etc.) and oxidants (e.g., $O_3$, $H_2O_2$ etc.). MADE/SORGAM scheme (Ackermann et al., 1998;Schell et al., 2001) with the improved ISORROPIA II model is used to simulate the aerosol dynamics (including nucleation, coagulation and condensation) and thermodynamics, and provide the aerosol size distribution, number concentration, as well as AWC and pH values. The Integrated process rate (IPR) technique (Tao et al., 2017;Tao et al., 2015) is used to record the formation rates of sulfuric and nitric acid vapor through photochemical oxidation
25    in the gas phase.

For the TMI oxidation pathway, we assume that $Fe^{3+}$ and $Mn^{2+}$ will not be consumed in the TMI pathway due to the catalytic nature of $Fe^{3+}/Mn^{2+}$. The concentrations of $Fe^{3+}$ and $Mn^{2+}$ (in unit of mol/L) in aerosol water can be calculated by Eq. (3),

$$\begin{cases} [\text{Fe}^{3+}] = Min(C_{\text{PM25\_FE}} \cdot FS_{\text{FE3+}} / AWC, 2.6 \times 10^{-38} / [\text{OH}^-]^3) \\ [\text{Mn}^{2+}] = Min(C_{\text{PM25\_MN}} \cdot FS_{\text{MN2+}} / AWC, 1.6 \times 10^{-13} / [\text{OH}^-]^2) \end{cases} \qquad (3)$$

where PM25_FE and PM25_MN are the fine particle components of Fe and Mn minerals, respectively, $C$ denotes the concentrations in unit of mol per liter of air, $FS_{\text{FE3+}}$ and $FS_{\text{MN2+}}$ represent the maximum fractional solubility of $\text{Fe}^{3+}$ and $\text{Mn}^{2+}$, respectively (regardless of the acidity of aerosol water), and $AWC$ is the aerosol liquid water content in unit of liter per liter of air.

**2.2 WRF-Chem model configuration and scenarios**

The modeling framework is constructed on a single domain of 100 (west-east) × 70 (south-north) × 30 (vertical layers) grid cells with a horizontal resolution of 20 km (including the Gobi deserts, see Fig. S8 of Supplement). The overview of the chemical and physical options used in this study is summarized in Table S6 of Supplement. Madronich F-TUV photolysis scheme (Madronich, 1987) is used to calculate the photolysis rates. The initial and boundary conditions for meteorology and chemistry are derived from 1.0°×1.0° NCEP FNL data and global-scale MOZART outputs, respectively. Observation nudging (Liu et al., 2005) is used to nudge the modeled temperature, wind fields and humidity towards the observations (including surface and upper layers). Multi-resolution Emission Inventory for China (MEIC) of the year 2013 (MEIC, Lei et al., 2011;Zhang et al., 2009;Li et al., 2014;Li et al., 2017b) is used for anthropogenic emissions. The Megan scheme (Guenther et al., 2006) is used for biogenic VOCs emissions. The hourly biomass burning emissions data are provided by the Fire Inventory from NCAR (FINN, Wiedinmyer et al., 2010). We use the GOCART dust scheme (Ginoux et al., 2001;Zhao et al., 2010;Zhao et al., 2013), which is coupled with the MADE/SORGAM aerosol scheme.

In this study, we have simulated 15 scenarios as detailed in Table 2, including an ORIG scenario with WRF-Chem (original chemistry) and default emissions as mentioned above, a CTRL scenario with WRF-Chem-AWAC (with implementation of aerosol water aqueous phase chemistry) and optimized ammonia emission, as well as additional emissions for chloride and crustal fine particles, and 13 sensitivity scenarios with respect to CTRL scenario. Since anthropogenic source chlorine is not included in MEIC, we adopt the chlorine inventory in Liu et al. (2018), which provides the emissions of HCl from coal consumption. WRF/Chem prescribes a specific mass ratio of the fine mode dust emission to the total dust emission, and we further adopt the fine mode dust emission speciation profiles from Dong et al. (2016). The mass fraction for fine particle components of K, Na, Ca, Mg, Fe and Mn minerals (denoted as PM25_K, PM25_NA, PM25_CA, PM25_MG, PM25_FE and PM25_MN, respectively) from dust source are set as 3.77%, 3.94%, 7.94%, 0.80%, 2.43%, and 0.063%, respectively. Dry and wet depositions are also considered for these newly-added crustal fine particle components as detailed in Sect. 1.3 of Supplement.

The CTRL scenario is expected to reproduce the observed fine particle compositions (including sulfate, nitrate, ammonium, chloride and crustal components) and gas phase pollutants, and thus more reliably predict the spatio-temporal distribution of pH, AWC and sulfate production. For this purpose, five parameters in the CTRL scenario have been adjusted to better match the observations (with the criteria that the relative error of the monthly mean concentrations for different fine particle components is less than 5%), namely the factor multiplied to the anthropogenic ammonia emissions (denoted as $emissf_{NH3}$), the factor multiplied to dust speciation fractions for PM25_K, PM25_NA, PM_CA and PM25_MG (denoted as $emissf_{OCAT}$), the factor multiplied to the anthropogenic chloride emissions (denoted as $emissf_{CL}$), as well as $FS_{FE3+}$ and $FS_{MN2+}$ (the maximum fractional solubility of $Fe^{3+}$ and $Mn^{2+}$ in Eq. 3, respectively). In the CTRL scenario, $emissf_{NH3}$ has been set to 2, as recent studies using top-down inverse modeling (Van Damme et al., 2018;Zhang et al., 2018a;Wang et al., 2018;Kong et al., 2019) and direct measurement (Wang et al., 2018) found that previous bottom-up inventories might underestimate the NH3 emissions, and MEIC inventory was estimated to under-predict NH3 emissions by about 40% over the North China Plain (Kong et al., 2019). As shown in Table S7 of Supplement, compared with the scenario using the default MEIC emission data, the CTRL scenario (with doubled NH3 emissions) better matches both the observed ammonia and ammonium concentrations at urban Beijing sites during wintertime (Meng et al., 2011;Liu et al., 2017;Song et al., 2018). To match the observations of PM25_OCAT, the $emissf_{OCAT}$ is set to 4.5 (the total dust emission is unchanged), and Dong et al. (2016) indicated that observed fine particles should have a considerably higher mass contribution within the East Asian dust than the chemical transport model prescribes. And $emissf_{CL}$ is set to 6 to match the observations of PM25_CL. To have a better agreement with the sulfate observation, $FS_{FE3+}$ and $FS_{MN2+}$ are set to 7% and 40%, respectively. Note that the assumption behind tuning only $FS_{FE3+}$ and $FS_{MN2+}$ to better agree with observed sulfates, is that the model could reasonably simulate the concentrations for other oxidants (e.g., OH, $H_2O_2$, $O_3$ and $NO_2$), thus the deviation from observation can be attributed to the uncertainties in representation of TMI pathway. Note that uncertainties in the emission, transport (i.e., advection and turbulent mixing), removal (dry and wet deposition) and sulfate formation in other phases could also contribute to the discrepancies between modeling results and observations. Nonetheless, this study does not aim at estimating the exact values for aerosol pH and sulfate formation budget . Instead, this study focuses on investigating the characterises in the spatio-temporal distribution for aerosol pH as well as sulfate formation budget, and also the uncertainties relevant with assumptions for input parameters. 
[revised manuscript text omitted]
). If PM25_OCAT emissions are removed or halved (OCAT0 and OCAT0.5 scenarios), averaged pH decreases to 4.2 and 4.6, respectively, and $P_{S(VI),surf}$ increases by ~250% and ~200%, respectively, compared to the CTRL scenario (a lower pH favors sulfate production through TMI pathway). Doubling both $NH_3$ and PM25_OCAT emissions (A2 and OCAT2 scenarios) leads to a slightly higher $pH_{surf}$ and a similar $P_{S(VI),surf}$, with the sulfate formation dominated by $NO_2$ and $O_3$ pathways. For the control experiments of MEIC_CTRL (using the original MEIC inventory), predicted $pH_{surf}$ is lower, and the relative contribution to sulfate formation is decreased for $NO_2$ pathway but increased for TMI pathway. Interestingly, rapid production of sulfate could be maintained over a wide pH range (~4.2-5.7) with the varying emissions Interestingly, rapid production of sulfate could be maintained over a wide pH range (~4.2-5.7) with the varying emissions Interestingly, rapid production of sulfate could be maintained over a wide pH range (~4.2-5.7) with the varying emissions Interestingly, rapid production of sulfate could be maintained over a wide pH range (~4.2-5.7) with the varying emissions for $NH_3$ and crustal particles (transition between TMI pathway dominated and $NO_2/O_3$ pathway dominated).

We have also investigated the effect of emissions of chlorides and $Fe^{3+}/Mn^{2+}$ ions. Removing all the chloride emissions (CL0 scenario) has a negligible effect on both aerosol pH and sulfate production. However, if the chloride emissions are

doubled (CL2 scenario), pH slightly decreases to 5.0, and $P_{S(VI),surf}$ increases by 50% (maybe increase in $NH_4Cl$ leading to an enhanced water absorption). When both $FS_{FE3+}$ and $FS_{MN2+}$ equal to zero (TMI0 scenario, and TMI pathway is shut down), $P_{S(VI),surf}$ decrease almost by half and $pH_{surf}$ (5.5) is slightly higher. Interestingly, when both $FS_{FE3+}$ and $FS_{MN2+}$ are halved (TMI0.5 scenario), a similar $pH_{surf}$ and $P_{S(VI),surf}$ is predicted as in TMI0 scenario. When both $FS_{FE3+}$ and

5 $FS_{MN2+}$ are doubled (TMI2 scenario), $P_{S(VI),surf}$ increase by ~300% and $pH_{surf}$ decreases to 4.6. Our results indicate that sulfate production is rather sensitive to the availability of TMI species. Unfortunately, the concentrations as well as sources for TMI species in aerosol water during haze episodes remain not well constrained and understood. The simulated mean concentration for $Fe^{3+}$ and $Mn^{2+}$ in $PM_{2.5}$ at the Beijing TSU site is 3.2 and 3.6 $ng/m^3$, respectively, and are smaller than the observed concentrations for soluble Fe and Mn (1.5-16 and 10-41 $ng/m^3$, respectively) in $PM_{2.5}$ at

10 a Xi'an site (Wang et al., 2016). Note that $Fe^{3+}/Mn^{2+}$ ions also have an anthropogenic source, and were estimated to account for 10-30% in Beijing (Shao et al., 2019). $Fe^{3+}/Mn^{2+}$ ions also have an anthropogenic source, and were estimated to account for 10-30% in Beijing (Shao et al., 2019). Furthermore, the soluble Fe/Mn speciation (including $Fe^{3+}$-$Fe^{2+}$, $Mn^{2+}$-$Mn^{3+}$-$Mn^{4+}$ cycling) depends on dust mineralogy, particle acidity and

[revised manuscript text omitted]

Pathways
$O_3$
$H_2O_2$
TMI
$NO_2$
$CH_3OOH$
GPP

Reference $P_{S(VI),surf}$
⊖ 0.5 µg/m³/h
⊖ 0.1 µg/m³/h

[revised manuscript text omitted]